# Estimating Mutual Information for Discrete-Continuous Mixtures

**Weihao Gao**
Department of ECE
Coordinated Science Laboratory
University of Illinois at Urbana-Champaign
wgao9@illinois.edu

**Sreeram Kannan**
Department of Electrical Engineering
University of Washington
ksreeram@uw.edu

**Sewoong Oh**
Department of IESE
Coordinated Science Laboratory
University of Illinois at Urbana-Champaign
swoh@illinois.edu

**Pramod Viswanath**
Department of ECE
Coordinated Science Laboratory
University of Illinois at Urbana-Champaign
pramodv@illinois.edu

## Abstract

Estimation of mutual information from observed samples is a basic primitive in machine learning, useful in several learning tasks including correlation mining, information bottleneck, Chow-Liu tree, and conditional independence testing in (causal) graphical models. While mutual information is a quantity well-defined for general probability spaces, estimators have been developed only in the special case of discrete or continuous pairs of random variables. Most of these estimators operate using the $3H$-principle, i.e., by calculating the three (differential) entropies of $X$, $Y$ and the pair $(X, Y)$. However, in general mixture spaces, such individual entropies are not well defined, even though mutual information is. In this paper, we develop a novel estimator for estimating mutual information in discrete-continuous mixtures. We prove the consistency of this estimator theoretically as well as demonstrate its excellent empirical performance. This problem is relevant in a wide-array of applications, where some variables are discrete, some continuous, and others are a mixture between continuous and discrete components.

## 1 Introduction

A fundamental quantity of interest in machine learning is mutual information (MI), which characterizes the shared information between a pair of random variables $(X, Y)$. MI obeys several intuitively appealing properties including the data-processing inequality, invariance under one-to-one transformations and chain rule [10]. Therefore, mutual information is widely used in machine learning for canonical tasks as classification [35], clustering [32, 49, 8, 29] and feature selection [2, 13]. Mutual information also emerges as the "correct" quantity in several graphical model inference problems (e.g., the Chow-Liu tree [9] and conditional independence testing [6]). MI is also pervasively used in many data science application domains, such as sociology [40], computational biology [28], and computational neuroscience [41].

An important problem in any of these applications is to estimate mutual information effectively from samples. While mutual information has been the *de facto* measure of information in several applications for decades, the estimation of mutual information from samples remains an active research problem. Recently, there has been a resurgence of interest in entropy, relative entropy and

mutual information estimators, on both the theoretical as well as practical fronts [46, 31, 44, 45, 22, 19, 7, 15, 14, 17, 16].

The previous estimators focus on either of two cases – the data is either purely discrete or purely continuous. In these special cases, the mutual information can be calculated based on the three (differential) entropies of $X$, $Y$ and $(X, Y)$. We term estimators based on this principle as $3H$-estimators (since they estimate three entropy terms), and a majority of previous estimators fall under this category [19, 16, 46].

In practical downstream applications, we often have to deal with a *mixture of continuous and discrete* random variables. Random variables can be mixed in several ways. First, one random variable can be discrete whereas the other is continuous. For example, we want to measure the strength of relationship between children's age and height, here age $X$ is discrete and height $Y$ is continuous. Secondly, a single scalar random variable itself can be a mixture of discrete and continuous components. For example, consider $X$ taking a zero-inflated-Gaussian distribution, which takes value 0 with probability 0.1 and is a Poisson distribution with mean 10 with probability 0.9. This distribution has both a discrete component as well as a component with density, and is a well-known model for gene expression readouts [24, 37]. Finally, $X$ and / or $Y$ can be high dimensional vector, each of whose components may be discrete, continuous or mixed.

In all of the aforementioned *mixed* cases, mutual information is well-defined through the Radon-Nikodym derivative (see Section 2) but cannot be expressed as a function of the entropies or differential entropies of the random variables. Crucially, entropy is not well defined when a single scalar random variable comprises of both discrete and continuous components, in which case, $3H$ estimators (the vast majority of prior art) cannot be directly employed. In this paper, we address this challenge by proposing an estimator that can handle all these cases of mixture distributions. The estimator directly estimates the Radon-Nikodym derivative using the $k$-nearest neighbor distances from the samples; we prove $\ell_2$ consistency of the estimator and demonstrate its excellent practical performance through a variety of experiments on both synthetic and real dataset. Most relevantly, it strongly outperforms natural baselines of discretizing the mixed random variables (by quantization) or making it continuous by adding a small Gaussian noise.

The rest of the paper is organized as follows. In Section 2, we review the general definition of mutual information for Radon-Nikodym derivative and show that it is well-defined for all the cases of mixtures. In Section 3, we propose our estimator of mutual information for mixed random variables. In Section 4, we prove that our estimator is $\ell_2$ consistent under certain technical assumptions and verify that the assumptions are satisfied for most practical cases. Section 5 contains the results of our detailed synthetic and real-world experiments testing the efficacy of the proposed estimator.

## 2 Problem Formation

In this section, we define mutual information for general distributions as follows (e.g., [39]).

**Definition 2.1.** *Let $P_{XY}$ be a probability measure on the space $\mathcal{X} \times \mathcal{Y}$, where $\mathcal{X}$ and $\mathcal{Y}$ are both Euclidean spaces. For any measurable set $A \subseteq \mathcal{X}$ and $B \subseteq \mathcal{Y}$, define $P_X(A) = P_{XY}(A \times \mathcal{Y})$ and $P_Y(B) = P_{XY}(\mathcal{X} \times B)$. Let $P_X P_Y$ be the product measure $P_X \times P_Y$. Then the mutual information $I(X; Y)$ of $P_{XY}$ is defined as*

$$I(X; Y) \equiv \int_{\mathcal{X} \times \mathcal{Y}} \log \frac{dP_{XY}}{dP_X P_Y} dP_{XY}, \tag{1}$$

*where $\frac{dP_{XY}}{dP_X P_Y}$ is the Radon-Nikodym derivative.*

We prove that for any probability measure $P$ on $\mathcal{X} \times \mathcal{Y}$, the joint measure $P_{XY}$ is absolutely continuous with respect to the product measure $P_X P_Y$, hence mutual information is well-defined. See Appendix **??** for the detailed proof. Notice that this general definition includes the following cases of mixtures: (1) $X$ is discrete and $Y$ is continuous (or vice versa); (2) $X$ or $Y$ has many components each, where some components are discrete and some are continuous; (3) $X$ or $Y$ or their joint distribution is a mixture of continuous and discrete distributions.

# 3 Estimators of Mutual Information

**Review of prior work**. The estimation problem is quite different depending on whether the underlying distribution is discrete, continuous or mixed. As pointed out earlier, most existing estimators for mutual information are based on the $3H$ principle: they estimate the three entropy terms first. This $3H$ principle can be applied only in the purely discrete or purely continuous case.

*Discrete data*: For entropy estimation of a discrete variable $X$, the straightforward approach to plug-in the estimated probabilities $\hat{p}_X(x)$ into the formula for entropy has been shown to be suboptimal [33, 1]. Novel entropy estimators with sub-linear sample complexity have been proposed [48, 53, 19, 21, 20, 23]. MI estimation can then be performed using the $3H$ principle, and such an approach is shown to be worst-case optimal for mutual-information estimation [19].

*Continuous data*: There are several estimators for differential entropy of continuous random variables, which have been exploited in a $3H$ principle to calculate the mutual information [3]. One family of entropy estimators are based on kernel density estimators [34] followed by re-substitution estimation. An alternate family of entropy estimators is based on $k$-Nearest Neighbor ($k$-NN) estimates, beginning with the pioneering work of Kozachenko and Leonenko [26] (the so-called KL estimator). Recent progress involves an inspired mixture of an ensemble of kernel and $k$-NN estimators [46, 4]. Exponential concentration bounds under certain conditions are in [43].

*Mixed Random Variables*: Since the entropies themselves may not be well defined for mixed random variables, there is no direct way to apply the $3H$ principle. However, once the data is quantized, this principle can be applied in the discrete domain. That mutual information in arbitrary measure spaces can indeed be computed as a maximum over quantization is a classical result [18, 36, 38]. However, the choice of quantization is complicated and while some quantization schemes are known to be consistent when there is a joint density[11], the mixed case is complex. Estimator of the average of Radon-Nikodym derivative has been studied in [50, 51]. Very recent work generalizing the ensemble entropy estimator when some components are discrete and others continuous is in [31].

*Beyond $3H$ estimation*: In an inspired work [27] proposed a *direct* method for estimating mutual information (KSG estimator) when the variables have a joint density. The estimator starts with the $3H$ estimator based on differential entropy estimates based on the $k$-NN estimates, and employs a heuristic to couple the estimates in order to improve the estimator. While the original paper did not contain any theoretical proof, even of consistency, its excellent practical performance has encouraged widespread adoption. Recent work [17] has established the consistency of this estimator along with its convergence rate. Further, recent works [14, 16] involving a combination of kernel density estimators and $k$-NN methods have been proposed to further improve the KSG estimator. [42] extends the KSG estimator to the case when one variable is discrete and another is scalar continuous.

None of these works consider a case even if one of the components has a mixture of continuous and discrete distribution, let alone for general probability distributions. There are two generic options: (1) one can add small independent noise on each sample to break the multiple samples and apply a continuous valued MI estimator (like KSG), or (2) quantize and apply discrete MI estimators but the performance for high-dimensional case is poor. These form baselines to compare against in our detailed simulations.

**Mixed Regime**. We first examine the behavior of other estimators in the mixed regime, before proceeding to develop our estimator. Let us consider the case when $X$ is discrete (but real valued) and $Y$ possesses a density. In this case, we will examine the consequence of using the $3H$ principle, with differential entropy estimated by the K-nearest neighbors. To do this, fix a parameter $k$, that determines the number of neighbors and let $\rho_{i,z}$ denote the distance of the $k$-the nearest neighbor of $z$, where $z = x$ or $z = y$ or $z = (x, y)$. Then $\widehat{I}_{3\mathrm{H}}^{(N)}(X; Y) =$

$$\left( \frac{1}{N} \sum_{i=1}^{N} \log \frac{N c_x \rho_{i,x}^d}{k} + a(k) \right) + \left( \frac{1}{N} \sum_{i=1}^{N} \log \frac{N c_y \rho_{i,y}^d}{k} + a(k) \right) - \left( \frac{1}{N} \sum_{i=1}^{N} \log \frac{N c_{xy} \rho_{i,xy}^d}{k} + a(k) \right)$$

where $\psi(\cdot)$ is the digamma function and $a(\cdot) = \log(\cdot) - \psi(\cdot)$. In the case that $X$ is discrete and $Y$ has a density, $I_{3\mathrm{H}}(X; Y) = -\infty + a - b = -\infty$, which is clearly wrong.

The basic idea of the KSG estimator is to ensure that the $\rho$ is the same for both $x$, $y$ and $(x, y)$ and the difference is instead in the number of nearest neighbors. Let $n_{x,i}$ be the number of samples of $X_i$'s

within distance $\rho_{i,xy}$ and $n_{y,i}$ be the number of samples of $Y_i$'s within distance $\rho_{i,xy}$. Then the KSG estimator is given by $\widehat{I}_{KSG}^{(N)} \equiv \frac{1}{N} \sum_{i=1}^{N} ( \psi(k) + \log(N) - \log(n_{x,i} + 1) - \log(n_{y,i} + 1) )$ where $\psi(\cdot)$ is the digamma function.

In the case of $X$ being discrete and $Y$ being continuous, it turns out that the KSG estimator does *not* blow up (unlike the $3H$ estimator), since the distances do not go to zero. However, in the mixed case, the estimator has a non-trivial bias due to discrete points and is no longer consistent.

**Proposed Estimator**. We propose the following estimator for general probability distributions, inspired by the KSG estimator. The intuition is as follows. Fist notice that MI is the average of the logarithm of Radon-Nikodym derivative, so we compute the Radon-Nikodym derivative for each sample $i$ and take the empirical average. The re-substitution estimator for MI is then given as follows: $\widehat{I}(X;Y) \equiv \frac{1}{n} \sum_{i=1}^{n} \log \left( \frac{dP_{XY}}{dP_X P_Y} \right)_{(x_i, y_i)}$. The basic idea behind our estimate of the Radon-Nikodym derivative at each sample point is as follows:

- When the point is discrete (which can be detected by checking if the $k$-nearest neighbor distance of data $i$ is zero), then we can assert that data $i$ is in a discrete component, and we can use plug-in estimator for Radon-Nikodym derivative.

- If the point is such that there is a joint density (locally), the KSG estimator suggests a natural idea: fix the radius and estimate the Radon-Nikodym derivative by $(\psi(k) + \log(N) - \log(n_{x,i} + 1) - \log(n_{y,i} + 1))$.

- If $k$-nearest neighbor distance is not zero, then it may be either purely continuous or mixed. But we show below that the method for purely continuous is also applicable for mixed.

Precisely, let $n_{x,i}$ be the number of samples of $X_i$'s within distance $\rho_{i,xy}$ and $n_{y,i}$ be the number of samples of $Y_i$'s with in $\rho_{i,xy}$. Denote $\tilde{k}_i$ by the number of tuples $(X_i, Y_i)$ within distance $\rho_{i,xy}$. If the $k$-NN distance is zero, which means that the sample $(X_i, Y_i)$ is a discrete point of the probability measure, we set $k$ to $\tilde{k}_i$, which is the number of samples that have the same value as $(X_i, Y_i)$. Otherwise we just keep $\tilde{k}_i$ as $k$. Our proposed estimator is described in detail in Algorithm 1.

---

**Algorithm 1** Mixed Random Variable Mutual Information Estimator

**Input:** $\{X_i, Y_i\}_{i=1}^{N}$, where $X_i \in \mathcal{X}$ and $Y_i \in \mathcal{Y}$;
**Parameter:** $k \in \mathbb{Z}^+$;
**for** $i = 1$ to $N$ **do**
    $\rho_{i,xy} :=$ the $k$ smallest distance among $[d_{i,j} := \max\{\|X_j - X_i\|, \|Y_j - Y_i\|\}, j \neq i]$;
    **if** $\rho_{i,xy} = 0$ **then**
        $\tilde{k}_i :=$ number of samples such that $d_{i,j} = 0$;
    **else**
        $\tilde{k}_i := k$;
    **end if**
    $n_{x,i} :=$ number of samples such that $\|X_j - X_i\| \leq \rho_{i,xy}$;
    $n_{y,i} :=$ number of samples such that $\|Y_j - Y_i\| \leq \rho_{i,xy}$;
    $\xi_i := \psi(\tilde{k}_i) + \log N - \log(n_{x,i} + 1) - \log(n_{y,i} + 1)$;
**end for**
**Output:** $\widehat{I}^{(N)}(X;Y) := \frac{1}{N} \sum_{i=1}^{N} \xi_i$.

---

We note that our estimator recovers previous ideas in several canonical settings. If the underlying distribution is discrete, the $k$-nearest neighbor distance $\rho_{i,xy}$ equals to 0 with high probability, then our estimator recovers the plug-in estimator. If the underlying distribution does not have probability masses, then there are no multiple overlapping samples, so $\tilde{k}_i$ equals to $k$, our estimator recovers the KSG estimator. If $X$ is discrete and $Y$ is single-dimensional continuous and $P_X(x) > 0$ for all $x$, for sufficiently large dataset, the $k$-nearest neighbors of sample $(x_i, y_i)$ will be located on the same $x_i$ with high probability. Therefore, our estimator recovers the discrete vs continuous estimator in [42].

# 4 Proof of Consistency

We show that under certain technical conditions on the joint probability measure, the proposed estimator is consistent. We begin with the following definitions.

$$P_{XY}(x, y, r) \equiv P_{XY}\left(\{(a, b) \in \mathcal{X} \times \mathcal{Y} : \|a - x\| \leq r, \|b - y\| \leq r\}\right), \tag{2}$$
$$P_X(x, r) \equiv P_X\left(\{a \in \mathcal{X} : \|a - x\| \leq r\}\right), \tag{3}$$
$$P_Y(y, r) \equiv P_Y\left(\{b \in \mathcal{Y} : \|b - y\| \leq r\}\right). \tag{4}$$

**Theorem 1.** *Suppose that*

1. *$k$ is chosen to be a function of $N$ such that $k_N \to \infty$ and $k_N \log N / N \to 0$ as $N \to \infty$.*

2. *The set of discrete points $\{(x, y) : P_{XY}(x, y, 0) > 0\}$ is finite.*

3. *$\frac{P_{XY}(x,y,r)}{P_X(x,r)P_Y(y,r)}$ converges to $f(x, y)$ as $r \to 0$ and $f(x, y) \leq C$ with probability 1.*

4. *$\mathcal{X} \times \mathcal{Y}$ can be decomposed into countable disjoint sets $\{E_i\}_{i=1}^{\infty}$ such that $f(x, y)$ is uniformly continuous on each $E_i$.*

5. *$\int_{\mathcal{X} \times \mathcal{Y}} \left| \log f(x, y) \right| dP_{XY} < +\infty.$*

*Then we have $\lim_{N \to \infty} \mathbb{E}\left[\widehat{I}^{(N)}(X; Y)\right] = I(X; Y)$.*

Notice that Assumptions 2,3,4 are satisfied whenever (1) the distribution is (finitely) discrete; (2) the distribution is continuous; (3) some dimensions are (countably) discrete and some dimensions are continuous; (4) a mixture of the previous cases. Most real world data can be covered by these cases. A sketch of the proof is below with the full proof in the supplementary material.

*Proof.* (Sketch) We start with an explicit form of the Radon-Nikodym derivative $dP_{XY}/(dP_X P_Y)$.

**Lemma 4.1.** *Under Assumption 3 and 4 in Theorem 1, $(dP_{XY}/(dP_X P_Y))(x, y) = f(x, y) = \lim_{r \to 0} P_{XY}(x, y, r)/(P_X(x, r)P_Y(y, r))$.*

Notice that $\widehat{I}_N(X; Y) = (1/N) \sum_{i=1}^{N} \xi_i$, where all $\xi_i$ are identically distributed. Therefore, $\mathbb{E}[\widehat{I}^{(N)}(X; Y)] = \mathbb{E}[\xi_1]$. Therefore, the bias can be written as:

$$\left| \mathbb{E}[\widehat{I}^{(N)}(X; Y)] - I(X; Y) \right| = \left| \mathbb{E}_{XY}\left[\mathbb{E}\left[\xi_1 | X, Y\right]\right] - \int \log f(X, Y) P_{XY} \right|$$

$$\leq \int \left| \mathbb{E}\left[\xi_1 | X, Y\right] - \log f(X, Y) \right| dP_{XY}. \tag{5}$$

Now we upper bound $\left| \mathbb{E}\left[\xi_1 | X, Y\right] - \log f(X, Y) \right|$ for every $(x, y) \in \mathcal{X} \times \mathcal{Y}$ by dividing the domain into three parts as $\mathcal{X} \times \mathcal{Y} = \Omega_1 \bigcup \Omega_2 \bigcup \Omega_3$ where

- $\Omega_1 = \{(x, y) : f(x, y) = 0\}$;

- $\Omega_2 = \{(x, y) : f(x, y) > 0, P_{XY}(x, y, 0) > 0\}$;

- $\Omega_3 = \{(x, y) : f(x, y) > 0, P_{XY}(x, y, 0) = 0\}$.

We show that $\lim_{N \to \infty} \int_{\Omega_i} \left| \mathbb{E}\left[\xi_1 | X, Y\right] - \log f(X, Y) \right| dP_{XY} = 0$ for each $i \in \{1, 2, 3\}$ separately.

- For $(x, y) \in \Omega_1$, we will show that $\Omega_1$ has zero probability with respect to $P_{XY}$, i.e. $P_{XY}(\Omega_1) = 0$. Hence, $\int_{\Omega_1} \left| \mathbb{E}\left[\xi_1 | X, Y\right] - \log f(X, Y) \right| dP_{XY} = 0$.

- For $(x, y) \in \Omega_2$, $f(x, y)$ equals to $P_{XY}(x, y, 0)/P_X(x, 0)P_Y(y, 0)$, so it can be viewed as a discrete part. We will first show that the $k$-nearest neighbor distance $\rho_{k,1} = 0$ with high probability. Then we will use the the number of samples on $(x, y)$ as $\tilde{k}_i$, and we will show that the mean of estimate $\xi_1$ is closed to $\log f(x, y)$.

- For $(x, y) \in \Omega_3$, it can be viewed as a continuous part. We use the similar proof technique as [27] to prove that the mean of estimate $\xi_1$ is closed to $\log f(x, y)$.

$\square$

The following theorem bounds the variance of the proposed estimator.

**Theorem 2.** *Assume in addition that*

   *6.* $(k_N \log N)^2/N \to 0$ *as* $N \to \infty$.

*Then we have*

$$\lim_{N \to \infty} \text{Var}\left[\widehat{I}^{(N)}(X; Y)\right] = 0 . \tag{6}$$

*Proof.* (Sketch) We use the Efron-Stein inequality to bound the variance of the estimator. For simplicity, let $\widehat{I}^{(N)}(Z)$ be the estimate based on original samples $\{Z_1, Z_2, \ldots, Z_N\}$, where $Z_i = (X_i, Y_i)$, and $\widehat{I}^{(N)}(Z_{\setminus j})$ is the estimate from $\{Z_1, \ldots, Z_{j-1}, Z_{j+1}, \ldots, Z_N\}$. Then a certain version of Efron-Stein inequality states that: $\text{Var}\left[\widehat{I}^{(N)}(Z)\right] \leq 2 \sum_{j=1}^{N} \left(\sup_{Z_1, \ldots, Z_N} \left|\widehat{I}^{(N)}(Z) - \widehat{I}^{(N)}(Z_{\setminus j})\right|\right)^2$.
Now recall that

$$\widehat{I}^{(N)}(Z) = \frac{1}{N} \sum_{i=1}^{N} \xi_i(Z) = \frac{1}{N} \sum_{i=1}^{N} \left(\psi(\tilde{k}_i) + \log N - \log(n_{x,i} + 1) - \log(n_{y,i} + 1)\right) , \tag{7}$$

Therefore, we have

$$\sup_{Z_1, \ldots, Z_N} \left|\widehat{I}^{(N)}(Z) - \widehat{I}^{(N)}(Z_{\setminus j})\right| \leq \frac{1}{N} \sup_{Z_1, \ldots, Z_N} \sum_{i=1}^{N} \left|\xi_i(Z) - \xi_i(Z_{\setminus j})\right| . \tag{8}$$

To upper bound the difference $|\xi_i(Z) - \xi_i(Z_{\setminus j})|$ created by eliminating sample $Z_j$ for different $i$ 's we consider three different cases: (1) $i = j$; (2) $\rho_{k,i} = 0$; (3) $\rho_{k,i} > 0$, and conclude that $\sum_{i=1}^{N} |\xi_i(Z) - \xi_i(Z_{\setminus j})| \leq O(k \log N)$ for all $Z_i$'s. The detail of the case study is in Section. **??** in the supplementary material. Plug it into Efron-Stein inequality, we obtain:

$$\text{Var}\left[\widehat{I}^{(N)}(Z)\right] \leq 2 \sum_{j=1}^{N} \left(\sup_{Z_1, \ldots, Z_N} \left|\widehat{I}^{(N)}(Z) - \widehat{I}^{(N)}(Z_{\setminus j})\right|\right)^2$$

$$\leq 2 \sum_{j=1}^{N} \left(\frac{1}{N} \sup_{Z_1, \ldots, Z_N} \sum_{i=1}^{N} \left|\xi_i(Z) - \xi_i(Z_{\setminus j})\right|\right)^2 = O((k \log N)^2/N) . \tag{9}$$

By Assumption 6, we have $\lim_{N \to \infty} \text{Var}\left[\widehat{I}^{(N)}(Z)\right] = 0$. $\square$

Combining Theorem 1 and Theorem 2, we have the $\ell_2$ consistency of $\widehat{I}^{(N)}(X; Y)$.

## 5  Simulations

We evaluate the performance of our estimator in a variety of (synthetic and real-world) experiments.

**Experiment I**. $(X, Y)$ is a mixture of one continuous distribution and one discrete distribution. The continuous distribution is jointly Gaussian with zero mean and covariance $\Sigma = \begin{pmatrix} 1 & 0.9 \\ 0.9 & 1 \end{pmatrix}$, and

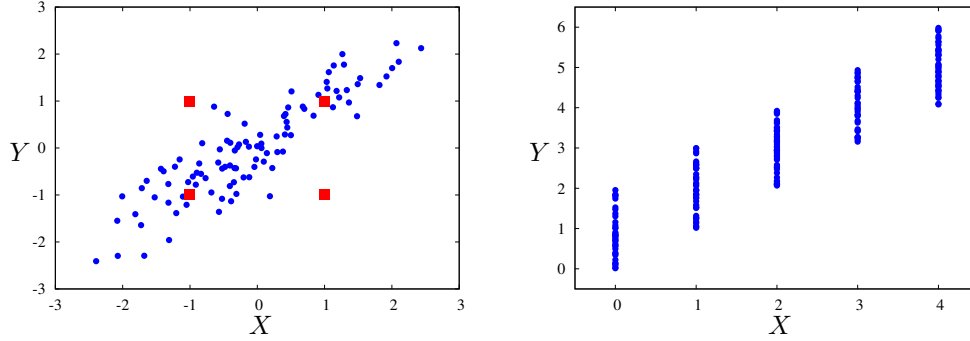

Figure 1: Left: An example of samples from a mixture of continuous (blue) and discrete (red) distributions, where red points denote multiple samples. Right: An example of samples from a discrete $X$ and a continuous $Y$.

the discrete distribution is $P(X = 1, Y = 1) = P(X = -1, Y = -1) = 0.45$ and $P(X = 1, Y = -1) = P(X = -1, 1) = 0.05$. These two distributions are mixed with equal probability. The scatter plot of a set of samples from this distribution is shown in the left panel of Figure. 1, where the red squares denote multiple samples from the discrete distribution. For all synthetic experiments, we compare our proposed estimator with a (fixed) partitioning estimator, an adaptive partitioning estimator [11] implemented by [47], the KSG estimator [27] and noisy KSG estimator (by adding Gaussian noise $N(0, \sigma^2 I)$ on each sample to transform all mixed distributions into continuous one). We plot the mean squared error versus number of samples in Figure 2. The mean squared error is averaged over 250 independent trials.

The KSG estimator is entirely misled by the discrete samples as expected. The noisy KSG estimator performs better but the added noise causes the estimate to degrade. In this experiment, the estimate is less sensitive to the noise added and the line is indistinguishable with the line for KSG. The partitioning and adaptive partitioning method quantizes all samples, resulting in an extra quantization error. Note that only the proposed estimator has error decreasing with the sample size.

**Experiment II**. $X$ is a discrete random variable and $Y$ is a continuous random variable. $X$ is uniformly distributed over integers $\{0, 1, \ldots, m-1\}$ and $Y$ is uniformly distributed over the range $[X, X + 2]$ for a given $X$. The ground truth $I(X; Y) = \log(m) - (m-1)\log(2)/m$. We choose $m = 5$ and a scatter plot of a set of samples is in the right panel of Figure. 1. Notice that in this case (and the following experiments) our proposed estimator degenerates to KSG if the hyper parameter $k$ is chosen the same, hence KSG is not plotted. In this experiment our proposed estimator outperforms other methods.

**Experiment III.** Higher dimensional mixture. Let $(X_1, Y_1)$ and $(Y_2, X_2)$ have the same joint distribution as in experiment II and independent of each other. We evaluate the mutual information between $X = (X_1, X_2)$ and $Y = (Y_1, Y_2)$. Then ground truth $I(X; Y) = 2(\log(m) - (m-1)\log(2)/m)$. We also consider $X = (X_1, X_2, X_3)$ and $Y = (Y_1, Y_2, Y_3)$ where $(X_3, Y_3)$ have the same joint distribution as in experiment II and independent of $(X_1, Y_1), (X_2, Y_2)$. The ground truth $I(X; Y) = 3(\log(m) - (m-1)\log(2)/m)$. The adaptive partitioning algorithm works only for one-dimensional $X$ and $Y$ and is not compared here.

We can see that the performance of partitioning estimator is very bad because the number of partitions grows exponentially with dimension. Proposed algorithm suffers less from the curse of dimensionality. For the right figure, noisy KSG method has smaller error, but we point out that it is unstable with respect to the noise level added: as the noise level is varied from $\sigma = 0.5$ to $\sigma = 0.7$ and the performance varies significantly (far from convergence).

**Experiment IV.** Zero-inflated Poissonization. Here $X \sim \text{Exp}(1)$ is a standard exponential random variable, and $Y$ is zero-inflated Poissonization of $X$, i.e., $Y = 0$ with probability $p$ and $Y \sim \text{Poisson}(x)$ given $X = x$ with probability $1 - p$. Here the ground truth is $I(X; Y) = (1 - p)(2\log 2 - \gamma - \sum_{k=1}^{\infty} \log k \cdot 2^{-k}) \approx (1-p)0.3012$, where $\gamma$ is Euler-Mascheroni constant. We repeat the experiment for no zero-inflation ($p = 0$) and for $p = 15\%$. We find that the proposed

estimator is comparable to adaptive partitioning for no zero-inflation and outperforms others for 15% zero-inflation.

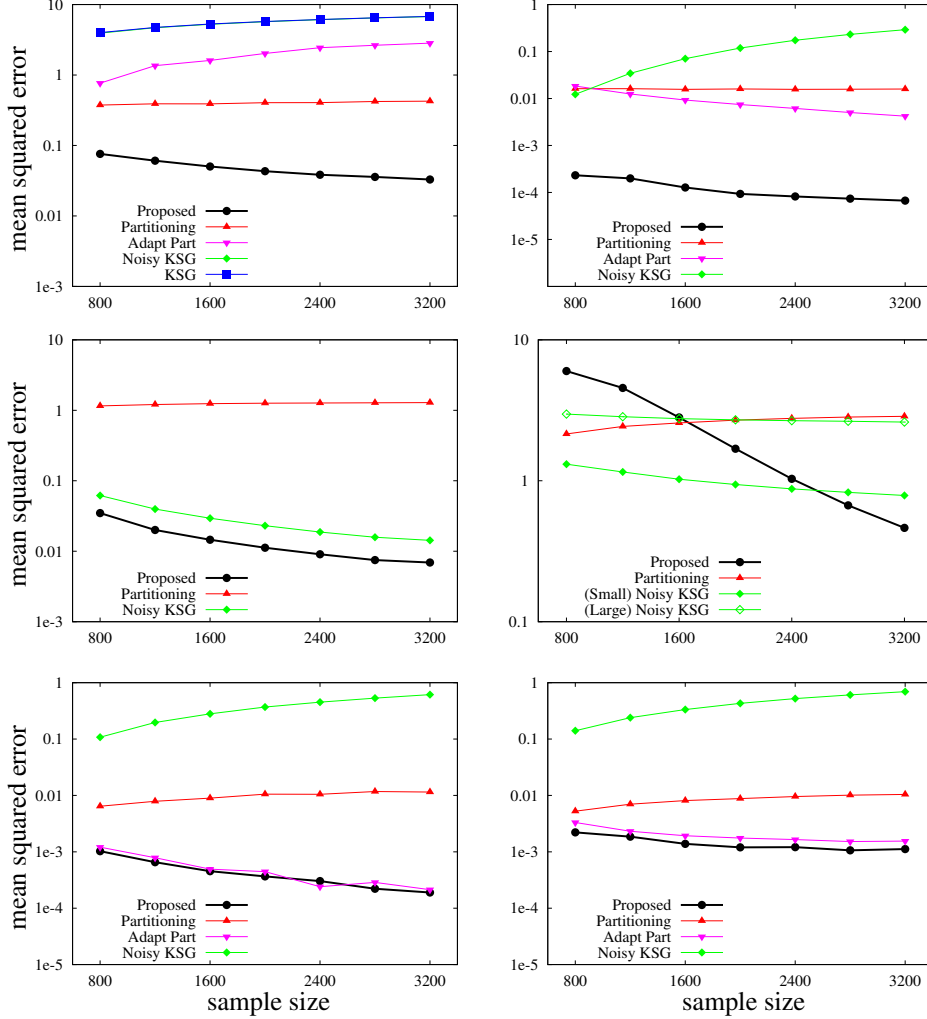

Figure 2: Mean squared error vs. sample size for synthetic experiments. Top row (left to right): Experiment I; Experiment II. Middle row (left to right): Experiment III for 4 dimensions and 6 dimensions. Bottom row (left to right): Experiment IV for $p = 0$ and $p = 15\%$.

We conclude that our proposed estimator is consistent for all these four experiments, and the mean squared error is always the best or comparable to the best. Other estimators are either not consistent or have large mean squared error for at least one experiment.

**Feature Selection Task**. Suppose there are a set of features modeled by independent random variables $(X_1, \ldots, X_p)$ and the data $Y$ depends on a subset of features $\{X_i\}_{i \in S}$, where $\mathrm{card}(S) = q < p$. We observe the features $(X_1, \ldots, X_p)$ and data $Y$ and try to select which features are related to $Y$. In many biological applications, some of the data is lost due to experimental reasons and set to 0; even the available data is noisy. This setting naturally leads to a mixture of continuous and discrete parts which we model by supposing that the observation is $\tilde{X}_i$ and $\tilde{Y}$, instead of $X_i$ and $Y$. Here $\tilde{X}_i$ and $\tilde{Y}$ equals to 0 with probability $\sigma$ and follows Poisson distribution parameterized by $X_i$ or $Y$ (which corresponds to the noisy observation) with probability $1 - \sigma$.

In this experiment, $(X_1, \ldots, X_{20})$ are i.i.d. standard exponential random variables and $Y$ is simply $(X_1, \ldots, X_5)$. $\tilde{X}_i$ equals to 0 with probability 0.15, and $\tilde{X}_i \sim \mathrm{Poisson}(X_i)$ with probability 0.85. $\tilde{Y}_i$ equals to 0 with probability 0.15 and $\tilde{Y}_i \sim \mathrm{Exp}(Y_i)$ with probability 0.85. Upon observing $\tilde{X}_i$'s

and $\tilde{Y}$, we evaluate $\text{MI}_i = I(\tilde{X}_i; Y)$ using different estimators, and select the features with top-$r$ highest mutual information. Since the underlying number of features is unknown, we iterate over all $r \in \{0, \ldots, p\}$ and observe a receiver operating characteristic (ROC) curve, shown in left of Figure 3. Compared to partitioning, noisy KSG and KSG estimators, we conclude that our proposed estimator outperforms other estimators.

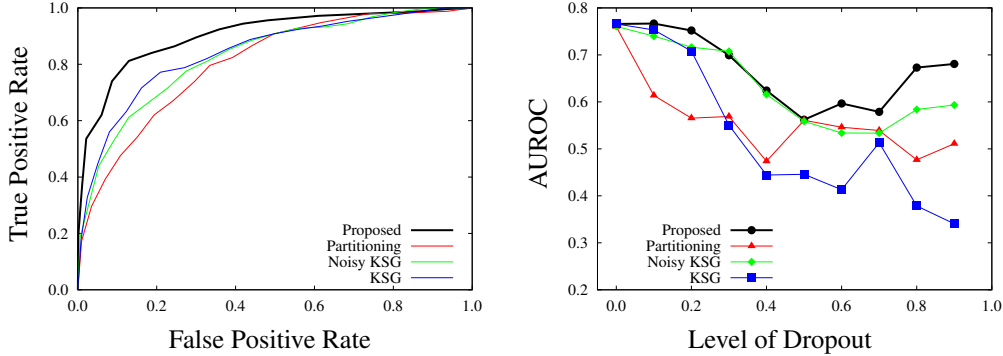

Figure 3: Left: ROC curve for the feature selection task. Right: AUROC versus levels of dropout for gene regulatory network inference.

**Gene regulatory network inference**. Gene expressions form a rich source of data from which to infer gene regulatory networks; it is now possible to sequence gene expression data from single cells using a technology called single-cell RNA-sequencing [52]. However, this technology has a problem called *dropout*, which implies that sometimes, even when the gene is present it is not sequenced [25, 12]. While we tested our algorithm on real single-cell RNA-seq dataset, it is hard to establish the ground truth on these datasets. Instead we resorted to a challenge dataset for reconstructing regulatory networks, called the DREAM5 challenge [30]. The simulated (insilico) version of this dataset contains gene expression for 20 genes with 660 data point containing various perturbations. The goal is to reconstruct the true network between the various genes. We used mutual information as the test statistic in order to obtain AUROC for various methods. While the dataset did not have any dropouts, in order to simulate the effect of dropouts in real data, we simulated various levels of dropout and compared the AUROC (area under ROC) of different algorithms in the right of Figure 3 where we find the proposed algorithm to outperform the competing ones.

## Acknowledgement

This work was partially supported by NSF grants CNS-1527754, CCF-1553452, CCF-1705007, CCF-1651236, CCF-1617745, CNS-1718270 and GOOGLE Faculty Research Award.

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
