[Supplementary Material]

# Appendix

## A Proof of Well-Definedness of Mutual Information

To prove the well-definedness of $I(X;Y)$, we need to show that $P_{XY}$ is absolutely continuous with respect to $P_X P_Y$. That is equivalent to show that for any measurable set $A \subseteq \mathcal{X} \times \mathcal{Y}$ such that $P_X P_Y(A) = 0$, we have $P_{XY}(A) = 0$. We will prove the contrapositive statement: for any measurable set $A \subseteq \mathcal{X} \times \mathcal{Y}$ such that $P_{XY}(A) > 0$, we have $P_X P_Y(A) > 0$. Consider a simple case that $A$ is a rectangle set, i.e. $A$ can be written as $A = A^x \times A^y$, where $A^x$, $A^y$ are measurable sets in $\mathcal{X}$ and $\mathcal{Y}$ respectively. Then

$$
\begin{aligned}
P_X P_Y(A) &= P_X(A^x) P_Y(A^y) = P_{XY}(A^x \times \mathcal{Y}) P_{XY}(\mathcal{X} \times A^y) \\
&\geq P_{XY}(A) P_{XY}(A) = (P_{XY}(A))^2 > 0
\end{aligned}
\tag{10}
$$

Since $\mathcal{X}$ and $\mathcal{Y}$ are Euclidean spaces, for any measurable set $A \subseteq \mathcal{X} \times \mathcal{Y}$, we can decompose $A$ as a countable union of disjoint rectangle sets. Let $A = \bigcup_{i=1}^{\infty} A_i$, where $A_i = A_i^x \times A_i^y$. Since $P_{XY}(A) > 0$, there exists $A_i$ such that $P_{XY}(A_i) > 0$, so $P_X P_Y(A_i) > 0$. Therefore, $P_X P_Y(A) > 0$.

Given that $P_{XY}$ is absolutely continuous with respect to $P_X P_Y$, by Radon-Nikodym theorem, there exists a function $f$ such that for any measurable set $A$, $\int_A f dP_X P_Y = P_{XY}(A)$. This $f$ is the Radon-Nikodym derivative $\frac{dP_{XY}}{dP_X P_Y}$ in (1).

## B Proof of Theorem 1

To prove the asymptotic unbiasedness of the estimator, we need to write the Radon-Nikodym derivative in an explicit form. The following lemma gives the explicit form of $\frac{dP_{XY}}{dP_X P_Y}$.

**Lemma B.1.** *Under Assumption 3 and 4 in Theorem 1, $\frac{dP_{XY}}{dP_X P_Y} = f(x,y) = \lim_{r \to 0} \frac{P_{XY}(x,y,r)}{P_X(x,r) P_Y(y,r)}$.*

Now notice that $\widehat{I}_N(X;Y) = \frac{1}{N} \sum_{i=1}^{N} \xi_i$, where all $\xi_i$ are identically distributed. Therefore, $\mathbb{E}[\widehat{I}_N(X;Y)] = \mathbb{E}[\xi_1]$. Therefore, the bias can be written as:

$$
\begin{aligned}
\left| \mathbb{E}[\widehat{I}_N(X;Y)] - I(X;Y) \right| &= \left| \mathbb{E}_{XY}\left[\mathbb{E}\left[\xi_1 | X, Y\right]\right] - \int \log f(X,Y) P_{XY} \right| \\
&\leq \int \left| \mathbb{E}\left[\xi_1 | X, Y\right] - \log f(X,Y) \right| dP_{XY}.
\end{aligned}
\tag{11}
$$

Now we will give upper bounds for $\left| \mathbb{E}\left[\xi_1 | X, Y\right] - \log f(X,Y) \right|$ for every $(x,y) \in \mathcal{X} \times \mathcal{Y}$. We will divide the space into three parts as $\mathcal{X} \times \mathcal{Y} = \Omega_1 \bigcup \Omega_2 \bigcup \Omega_3$ where

- $\Omega_1 = \{(x,y) : f(x,y) = 0\}$;
- $\Omega_2 = \{(x,y) : f(x,y) > 0, P_{XY}(x,y,0) > 0\}$;
- $\Omega_3 = \{(x,y) : f(x,y) > 0, P_{XY}(x,y,0) = 0\}$.

We will show that $\lim_{N \to \infty} \int_{\Omega_i} \left| \mathbb{E}\left[\xi_1 | (X,Y) = (x,y)\right] - \log f(x,y) \right| dP_{XY} = 0$ for each $i \in \{1,2,3\}$ separately.

$(x,y) \in \Omega_1$: In this case, we will show that $\Omega_1$ has zero probability with respect to $P_{XY}$.

$$
P_{XY}(\Omega_1) = \int_{\Omega_1} dP_{XY} = \int_{\Omega_1} f(X,Y) dP_X P_Y = \int_{\Omega_1} 0 \, dP_X P_Y = 0
\tag{12}
$$

Therefore, $\int_{\Omega_1} \left| \mathbb{E}\left[\xi_1 | X, Y\right] - \log f(X,Y) \right| dP_{XY} = 0$.

$(x, y) \in \Omega_2$: In this case, $f(x, y)$ is just $P_{XY}(x, y, 0)/P_X(x, 0)P_Y(y, 0)$. We will first show that the probability that the $k$-nearest neighbor distance $\rho_{k,1} > 0$ is small. Then with high probability, we will use the the number of samples on $(x, y)$ as $\tilde{k}_i$, and we will show that the mean of estimate $\xi_1$ is closed to $\log f(x, y)$.

First, the probability of $\rho_{k,1} > 0$ is upper bounded by:

$$\mathbb{P}\left(\rho_{k,1} > 0 \,|\, (X, Y) = (x, y)\right)$$

$$= \sum_{m=0}^{k-1} \binom{N-1}{m} P_{XY}(x, y, 0)^m (1 - P_{XY}(x, y, 0))^{N-1-m}$$

$$\leq \sum_{m=0}^{k-1} N^m (1 - P_{XY}(x, y, 0))^{N-k}$$

$$\leq kN^k (1 - P_{XY}(x, y, 0))^{N-k}$$

$$\leq kN^k e^{-(N-k)P_{XY}(x,y,0)} . \tag{13}$$

Conditioning on the event that $\rho_{k,1} = 0$, we have $\xi_1 = \psi(\tilde{k}_1) + \log N - \log(n_{x,1} + 1) - \log(n_{y,1} + 1)$, where the distribution of $\tilde{k}_1$, $n_{x,1}$ and $n_{y,1}$ are given by the following lemma.

**Lemma B.2.** *Given $(X, Y) = (x, y)$ and $\rho_{k,1} = 0$, then $\tilde{k}_1 - k$ is distributed as $\mathrm{Bino}(N - k - 1, P_{XY}(x, y, 0))$; $n_{x,1} - k$ is distributed as $\mathrm{Bino}(N - k - 1, P_X(x, 0))$; $n_{y,1} - k$ is distributed as $\mathrm{Bino}(N - k - 1, P_Y(y, 0))$. Given $(X, Y) = (x, y)$ and $\rho_{k,1} = r > 0$, then $n_{x,1} - k$ is distributed as $\mathrm{Bino}(N - k - 1, \frac{P_X(x,r) - P_{XY}(x,y,r)}{1 - P_{XY}(x,y,r)})$; $n_{y,1} - k$ is distributed as $\mathrm{Bino}(N - k - 1, \frac{P_Y(y,r) - P_{XY}(x,y,r)}{1 - P_{XY}(x,y,r)})$.*

Then we write $\left| \mathbb{E}\left[\xi_1 | (X, Y) = (x, y), \rho_{k,1} = 0\right] - \log f(x, y) \right|$ as

$$\left| \mathbb{E}\left[\xi_1 | (X, Y) = (x, y), \rho_{k,1} = 0\right] - \log f(x, y) \right|$$

$$= \left| \mathbb{E}\left[\psi(\tilde{k}_1) + \log N - \log(n_{x,1} + 1) - \log(n_{y,1} + 1)|(X, Y) = (x, y), \rho_{k,1} = 0\right] \right.$$

$$\left. - \log \frac{P_{XY}(x, y, 0)}{P_X(x, 0)P_Y(y, 0)} \right|$$

$$\leq \left| \mathbb{E}\left[\log(n_{x,1} + 1)|(X, Y) = (x, y), \rho_{k,1} = 0\right] - \log NP_X(x, 0) \right|$$

$$+ \left| \mathbb{E}\left[\log(n_{y,1} + 1)|(X, Y) = (x, y), \rho_{k,1} = 0\right] - \log NP_Y(y, 0) \right|$$

$$+ \left| \mathbb{E}\left[\psi(\tilde{k}_1)|(X, Y) = (x, y), \rho_{k,1} = 0\right] - \log NP_{XY}(x, y, 0) \right| \tag{14}$$

By Lemma B.2, we know that $n_{x,i} - k$ is distributed as $\mathrm{Bino}(N - k - 1, P_X(x, 0))$. The following lemma establishes the mean of $\log(n_{x,i} + 1)$.

**Lemma B.3.** *If $X$ is distributed as $\mathrm{Bino}(N, p)$, then $|\mathbb{E}[\log(X + k)] - \log(Np + k)| \leq C/(Np + k)$ for some constant $C$.*

Therefore, the first term of (14) is bounded by:

$$\left| \mathbb{E}\left[\log(n_{x,1} + 1)|(X, Y) = (x, y), \rho_{k,1} = 0\right] - \log NP_X(x, 0) \right|$$

$$\leq \left| \mathbb{E}\left[\log(n_{x,1} + 1)|(X, Y) = (x, y), \rho_{k,1} = 0\right] - \log((N - k - 1)P_X(x, 0) + k + 1) \right|$$

$$+ \left| \log((N - k - 1)P_X(x, 0) + k + 1) - \log NP_X(x, 0) \right|$$

$$\leq \frac{C}{(N - k - 1)P_X(x, 0) + k + 1} + \left| \log \frac{(N - k - 1)P_X(x, 0) + k + 1}{NP_X(x, 0)} \right|$$

$$\leq \frac{C}{NP_X(x, 0)} + \log(1 + \frac{(k + 1)(1 - P_X(x, 0))}{NP_X(x, 0)})$$

$$\leq \frac{C}{NP_X(x, 0)} + \frac{(k + 1)(1 - P_X(x, 0))}{NP_X(x, 0)} \leq \frac{k + C + 1}{NP_X(x, 0)} . \tag{15}$$

where we use the fact that $\log(1+x) < x$ for all $x > 0$. Similarly, the second term of (14) is bounded by: $(k + C + 1)/(NP_Y(y,0))$. For the third term, notice that $|\psi(x) - \log(x)| \leq 1/x$ for every integer $x \geq 1$, therefore, $|\psi(\tilde{k}_1) - \log(\tilde{k}_1)| \leq 1/\tilde{k}_1 \leq 1/k$. So the third term of (14) is bounded by: $(k + C + 1)/(NP_{XY}(x,y,0)) + 1/k$. By Combining three terms together and noticing that $P_X(x,0) \geq P_{XY}(x,y,0)$ and $P_Y(y,0) \geq P_{XY}(x,y,0)$, we obtain

$$\left| \mathbb{E}\left[\xi_1 | (X,Y) = (x,y), \rho_{k,1} = 0\right] - \log f(x,y) \right|$$
$$\leq \frac{k+C+1}{NP_X(x,0)} + \frac{k+C+1}{NP_Y(y,0)} + \frac{k+C+1}{NP_{XY}(x,y,0)} + \frac{1}{k} \leq \frac{3k+3C+3}{NP_{XY}(x,y,0)} + \frac{1}{k} . \quad (16)$$

Combine with the case that $\rho_{i,xy} > 0$, we obtain that:

$$\left| \mathbb{E}\left[\xi_1 | (X,Y) = (x,y)\right] - \log f(x,y) \right|$$
$$\leq \left| \mathbb{E}\left[\xi_1 | (X,Y) = (x,y), \rho_{k,1} > 0\right] - \log f(x,y) \right| \times \mathbb{P}\left(\rho_{k,1} > 0\right)$$
$$+ \left| \mathbb{E}\left[\xi_1 | (X,Y) = (x,y), \rho_{k,1} = 0\right] - \log f(x,y) \right| \times \mathbb{P}\left(\rho_{k,1} = 0\right)$$
$$\leq (2\log N + |\log f(x,y)|)kN^k e^{-(N-k)P_{XY}(x,y,0)} + \frac{3k+3C+3}{NP_{XY}(x,y,0)} + \frac{1}{k} , \quad (17)$$

where the first term comes from triangle inequality and the fact that $|\xi_1| \leq 2\log N$. Integrating over $\Omega_2$, we have:

$$\int_{\Omega_2} \left| \mathbb{E}\left[\xi_1 | (X,Y) = (x,y)\right] - \log f(x,y) \right| dP_{XY}$$
$$\leq \int_{\Omega_2} (2\log N + |\log f(x,y)|)kN^k e^{-(N-k)P_{XY}(x,y,0)} dP_{XY}$$
$$+ \frac{3k+3C+3}{N} \int_{\Omega_2} \frac{1}{P_{XY}(x,y,0)} dP_{XY} + \frac{1}{k}$$
$$\leq (2\log N + \int_{\Omega_2} |\log f(x,y)| dP_{XY})kN^k e^{-(N-k)\inf_{(x,y)\in\Omega_2} P_{XY}(x,y,0)}$$
$$+ \frac{3k+3C+3}{N} \mu(\Omega_2) + \frac{1}{k} , \quad (18)$$

where $\mu$ denotes counting measure. By Assumption 1, $k$ goes to infinity as $N$ goes to infinity, so $1/k$ vanishes as $N$ increases. By Assumption 1 and 2, $k/N$ goes to 0 and $\Omega_2$ has finite counting measure, so the second term also vanishes. Since $\Omega_2$ has finite counting measure, so $\inf_{(x,y)\in\Omega_2} P_{XY}(x,y,0) = \epsilon > 0$. By Assumption 5, $\int_{\Omega_2} |\log f(x,y)| dP_{XY} < +\infty$. Therefore, for sufficiently large $N$, the first term also vanishes. Therefore,

$$\lim_{N\to\infty} \int_{\Omega_2} \left| \mathbb{E}\left[\xi_1 | (X,Y) = (x,y)\right] - \log f(x,y) \right| dP_{XY} = 0 . \quad (19)$$

$(x,y) \in \Omega_3$: In this case, $P_{XY}(x,y,r)$ is a monotonic function of $r$ such that $P_{XY}(x,y,0) = 0$ and $\lim_{r\to\infty} P_{XY}(x,y,r) = 1$. Hence, we can view $\log\left(P_{XY}(x,y,r)/P_X(x,r)P_Y(y,r)\right)$ as a function of $P_{XY}(x,y,r)$, and it converges to $\log f(x,y)$ as $P_{XY}(x,y,r) \to 0$, for almost every $(x,y)$. Since $P_{XY}(\Omega_3) \leq 1 < +\infty$ and $\int_{\Omega_3} |\log f(x,y)| dP_{XY} < +\infty$. Then by Egoroff's Theorem, for any $\epsilon > 0$, there exists a subset $E \subseteq \Omega_3$ with $P_{XY}(E) < \epsilon$ and $\int_E |\log f(x,y)| dP_{XY} < \epsilon$, such that $\log\left(P_{XY}(x,y,r)/P_X(x,r)P_Y(y,r)\right)$ converges as $P_{XY}(x,y,r) \to 0$, uniformly on $\Omega_3 \setminus E$. For $(x,y) \in E$, notice that $|\xi_1| \leq 2\log N$, so we have:

$$\int_E \left| \mathbb{E}\left[\xi_1 | (X,Y) = (x,y)\right] - \log f(x,y) \right| dP_{XY}$$
$$\leq \int_E \left(2\log N + |\log f(x,y)|\right) dP_{XY} < (2\log N + 1)\epsilon . \quad (20)$$

By choosing $\epsilon$ appropriately, we will have $\lim_{N\to\infty} \int_E \left| \mathbb{E}\left[\xi_1 | (X,Y) = (x,y)\right] - \log f(x,y) \right| dP_{XY} = 0$.

Now for any $(x,y) \in \Omega_3 \setminus E$, since $P_{XY}(x,y,0) = 0$, we know that $\mathbb{P}(\rho_{k,1} = 0 \mid (X,Y) = (x,y)) = 0$, so $\tilde{k}_1 = k$ with probability 1. Conditioning on $\rho_{k,1} = r > 0$, the difference $\left| \mathbb{E}[\xi_1 | (X,Y) = (x,y)] - \log f(x,y) \right|$ can be decomposed into four parts as follows

$$
\left| \mathbb{E}[\xi_1 | (X,Y) = (x,y)] - \log f(x,y) \right|
$$

$$
= \left| \int_{r=0}^{\infty} \left( \mathbb{E}[\xi_1 | (X,Y) = (x,y), \rho_{k,1} = r] - \log f(x,y) \right) dF_{\rho_{k,1}}(r) \right|
$$

$$
\leq \left| \int_{r=0}^{\infty} \left( \log \frac{P_{XY}(x,y,r)}{P_X(x,r) P_Y(y,r)} - \log f(x,y) \right) dF_{\rho_{k,1}}(r) \right| \tag{21}
$$

$$
+ \left| \int_{r=0}^{\infty} \left( \psi(k) - \log N - \log P_{XY}(x,y,r) \right) dF_{\rho_{k,1}}(r) \right| \tag{22}
$$

$$
+ \left| \int_{r=0}^{\infty} \left( \mathbb{E}[\log(n_{x,1}+1) | (X,Y,\rho_{k,1}) = (x,y,r)] - \log(N P_X(x,r)) \right) dF_{\rho_{k,1}}(r) \right| \tag{23}
$$

$$
+ \left| \int_{r=0}^{\infty} \left( \mathbb{E}[\log(n_{y,1}+1) | (X,Y,\rho_{k,1}) = (x,y,r)] - \log(N P_Y(y,r)) \right) dF_{\rho_{k,1}}(r) \right| \tag{24}
$$

here $F_{\rho_{k,1}}(r)$ is the CDF of the $k$-nearest neighbor distance $\rho_{k,1}$, given $(X,Y) = (x,y)$. By results of order statistics, its derivative with respect to $P_{XY}(x,y,r)$ is given by:

$$
\frac{dF_{\rho_{k,1}}(r)}{dP_{XY}(x,y,r)} = \frac{(N-1)!}{(k-1)!(N-k-1)!} P_{XY}(x,y,r)^{k-1} \left( 1 - P_{XY}(x,y,r) \right)^{N-k-1} . \tag{25}
$$

Now we consider the four terms separately. For (21), since $\log\left( P_{XY}(x,y,r)/P_X(x,r)P_Y(y,r) \right)$ converges as $P_{XY}(x,y,r) \to 0$, uniformly on $\Omega_3 \setminus E$. So for every $(x,y) \in \Omega_3 \setminus E$, there exists an $r_N$ such that $P_{XY}(x,y,r_N) = 4k \log N / N$ and $\left| \log\left( P_{XY}(x,y,r)/P_X(x,r)P_Y(y,r) \right) - \log f(x,y) \right| < \delta_N$ for every $r \leq r_N$. Here $r_N$ may depend on $(x,y)$, but $\delta_N$ does not depend on $(x,y)$ and $\lim_{N \to \infty} \delta_N = 0$. Therefore, (21) is upper bounded by:

$$
\left| \int_{r=0}^{\infty} \left( \log \frac{P_{XY}(x,y,r)}{P_X(x,r) P_Y(y,r)} - \log f(x,y) \right) dF_{\rho_{k,1}}(r) \right|
$$

$$
\leq \int_{r=0}^{r_N} \left| \log \frac{P_{XY}(x,y,r)}{P_X(x,r) P_Y(y,r)} - \log f(x,y) \right| dF_{\rho_{k,1}}(r)
$$

$$
+ \int_{r=r_N}^{\infty} \left| \log \frac{P_{XY}(x,y,r)}{P_X(x,r) P_Y(y,r)} - \log f(x,y) \right| dF_{\rho_{k,1}}(r)
$$

$$
\leq \delta_N \mathbb{P}(\rho_{k,1} \leq r_N \mid (X,Y) = (x,y))
$$

$$
+ \left( \sup_{r \geq r_N} \left| \log \frac{P_{XY}(x,y,r)}{P_X(x,r) P_Y(y,r)} - \log f(x,y) \right| \right) \mathbb{P}(\rho_{k,1} > r_N \mid (X,Y) = (x,y)) \tag{26}
$$

Firstly, the probability $\mathbb{P}(\rho_{k,1} \leq r_N \mid (X,Y) = (x,y))$ is smaller than 1. Secondly, since $P_X(x,y,r) \geq 4k \log N / N > 1/N$ for $r \geq r_N$, so we have $|\log P_{XY}(x,y,r)| \leq \log N$. The same bounds apply for $|\log P_X(x,r)|$ and $|\log P_Y(y,r)|$ as well. By triangle inequality, the supremum is upper bounded by $3 \log N + |\log f(x,y)|$. Finally, the probability $\mathbb{P}(\rho_{k,1} > r_N \mid (X,Y) = (x,y))$ is upper bounded by

$$
\mathbb{P}(\rho_{k,1} > r_N \mid (X,Y) = (x,y))
$$

$$
= \sum_{m=0}^{k-1} \binom{N-1}{m} P_{XY}(x,y,r_N)^m (1 - P_{XY}(x,y,r_N))^{N-1-m}
$$

$$
\leq \sum_{m=0}^{k-1} N^m (1 - P_{XY}(x,y,r_N))^{N-k}
$$

$$
= kN^k \left( 1 - \frac{4k \log N}{N} \right)^{N/2}
$$

$$
\leq kN^k e^{-2k \log N} = \frac{k}{N^k} . \tag{27}
$$

for sufficiently large $N$ such that $N - k > N/2$. Therefore, (21) is upper bounded by

$$\left| \int_{r=0}^{\infty} \left( \log \frac{P_{XY}(x,y,r)}{P_X(x,r)P_Y(y,r)} - \log f(x,y) \right) dF_{\rho_{k,1}}(r) \right|$$
$$\leq \quad \delta_N + \frac{k(3\log N + |\log f(x,y)|)}{N^k} \ . \tag{28}$$

For (22), we simply plug in $F_{\rho_{k,1}}(r)$ and integrate over $P_{XY}(x,y,r)$ and obtain

$$\int_{r=0}^{\infty} \left( \psi(k) - \log N - \log P_{XY}(x,y,r) \right) dF_{\rho_{k,1}}(r)$$
$$= \quad \psi(k) - \log N - \frac{(N-1)!}{(k-1)!(N-k-1)!}$$
$$\times \int_{r=0}^{\infty} (\log P_{XY}(x,y,r)) P_{XY}(x,y,r)^{k-1} \left( 1 - P_{XY}(x,y,r) \right)^{N-k-1} dP_{XY}(x,y,r)$$
$$= \quad \psi(k) - \log N - \frac{(N-1)!}{(k-1)!(N-k-1)!} \int_{t=0}^{1} (\log t) t^{k-1} (1-t)^{N-k-1} dt$$
$$= \quad \psi(k) - \log N - (\psi(k) - \psi(N)) = \psi(N) - \log N \ . \tag{29}$$

where we use the fact that $\psi(k) - \psi(N) = \frac{(N-1)!}{(k-1)!(N-k-1)!} \int_{t=0}^{1} (\log t) t^{k-1} (1-t)^{N-k-1} dt$. Notice that $\psi(N) < \log N$ and $\lim_{N \to 0} (\psi(N) - \log N) = 0$.

For (23), recall that in Lemma B.2, we have shown that conditioning on $(X,Y) = (x,y)$ and $\rho_{k,1} = r > 0$, $n_{x,1} - k$ is distributed as $\text{Bino}(N - k - 1, (P_X(x,r) - P_{XY}(x,y,r))/(1 - P_{XY}(x,y,r)))$. The expectation $\mathbb{E}\left[\log(n_{x,1} + 1)|(X,Y) = (x,y), \rho_{k,1} = r\right]$ is given by Lemma B.3. Therefore, we can rewrite the term (23) as:

$$\left| \int_{r=0}^{\infty} \left( \mathbb{E}\left[\log(n_{x,1} + 1)|(X,Y) = (x,y), \rho_{k,1} = r\right] - \log N - \log P_X(x,r) \right) dF_{\rho_{k,1}}(r) \right|$$
$$\leq \quad \left| \int_{r=0}^{\infty} \left( \mathbb{E}\left[\log(n_{x,1} + 1)|(X,Y) = (x,y), \rho_{k,1} = r\right] \right. \right.$$
$$\left. \left. - \log \left( (N - k - 1)\frac{P_X(x,r) - P_{XY}(x,y,r)}{1 - P_{XY}(x,y,r)} + k + 1 \right) \right) dF_{\rho_{k,1}}(r) \right|$$
$$+ \left| \int_{r=0}^{\infty} \left( \log \frac{(N - k - 1)\frac{P_X(x,r) - P_{XY}(x,y,r)}{1 - P_{XY}(x,y,r)} + k + 1}{N P_X(x,r)} \right) dF_{\rho_{k,1}}(r) \right|$$
$$\leq \quad \int_{r=0}^{\infty} \left| \mathbb{E}\left[\log(n_{x,1} + 1)|(X,Y) = (x,y), \rho_{k,1} = r\right] \right.$$
$$\left. - \log \left( (N - k - 1)\frac{P_X(x,r) - P_{XY}(x,y,r)}{1 - P_{XY}(x,y,r)} + k + 1 \right) \right| dF_{\rho_{k,1}}(r) \tag{30}$$
$$+ \left| \mathbb{E}_r \left[ \log \left( \frac{N(P_X(x,r) - P_{XY}(x,y,r)) + (k+1)(1 - P_X(x,r))}{N P_X(x,r)(1 - P_{XY}(x,y,r))} \right) \right] \right| \ . \tag{31}$$

where $\mathbb{E}_r$ denotes expectation over $F_{\rho_{i,xy}}$. By Lemma B.3, the term (30) is upper bounded by

$$\int_{r=0}^{\infty} \left| \mathbb{E}\left[\log(n_{x,1} + 1)|(X,Y) = (x,y), \rho_{k,1} = r\right] \right.$$
$$\left. - \log \left( (N - k - 1)\frac{P_X(x,r) - P_{XY}(x,y,r)}{1 - P_{XY}(x,y,r)} + k + 1 \right) \right| dF_{\rho_{k,1}}(r)$$
$$\leq \quad \int_{r=0}^{\infty} \frac{C}{(N - k - 1)\frac{P_X(x,r) - P_{XY}(x,y,r)}{1 - P_{XY}(x,y,r)} + k + 1} dF_{\rho_{k,1}}(r)$$
$$\leq \quad \int_{r=0}^{\infty} \frac{C}{k + 1} dF_{\rho_{k,1}}(r) = \frac{C}{k + 1} \ . \tag{32}$$

For (31), by the fact that $\log(x/y) \leq (x - y)/y$ for all $x, y > 0$ and Cauchy-Schwarz inequality, we have the following:

$$
\mathbb{E}_r \left[ \log \left( \frac{N(P_X(x,r) - P_{XY}(x,y,r)) + (k+1)(1 - P_X(x,r))}{NP_X(x,r)(1 - P_{XY}(x,y,r))} \right) \right]
$$

$$
\leq \quad \mathbb{E}_r \left[ \frac{N(P_X(x,r) - P_{XY}(x,y,r)) + (k+1)(1 - P_X(x,r))}{NP_X(x,r)(1 - P_{XY}(x,y,r))} - 1 \right]
$$

$$
= \quad \mathbb{E}_r \left[ \frac{(k + 1 - NP_{XY}(x,y,r))(1 - P_X(x,r))}{NP_X(x,r)(1 - P_{XY}(x,y,r))} \right]
$$

$$
\leq \quad \sqrt{\mathbb{E}_r \left[ \left( \frac{k + 1 - NP_{XY}(x,y,r)}{NP_{XY}(x,y,r)} \right)^2 \right] \mathbb{E}_r \left[ \left( \frac{P_{XY}(x,y,r)(1 - P_X(x,r))}{P_X(x,r)(1 - P_{XY}(x,y,r))} \right)^2 \right]} . \quad (33)
$$

Notice that $P_X(x,r) \geq P_{XY}(x,y,r)$ for all $r$, so the second expectation is always no larger than 1. For the first expectation, we plug in $F_{\rho_{k,1}}(r)$ and integrate over $P_{XY}(x,y,r)$, let $t = P_{XY}(x,y,r)$ and observe,

$$
\mathbb{E}_r \left[ \left( \frac{k + 1 - NP_{XY}(x,y,r)}{NP_{XY}(x,y,r)} \right)^2 \right]
$$

$$
= \quad \int_{r=0}^{\infty} \left( \frac{k + 1 - NP_{XY}(x,y,r)}{NP_{XY}(x,y,r)} \right)^2 dF_{\rho_{i,xy}}(r)
$$

$$
= \quad \frac{(N-1)!}{(k-1)!(N-k-1)!} \int_{t=0}^{1} \frac{(k+1-Nt)^2}{N^2 t^2} t^{k-1}(1-t)^{N-k-1} dt
$$

$$
= \quad \frac{(N-1)!}{(k-1)!(N-k-1)!} \frac{(k+1)^2}{N^2} \int_{t=0}^{1} t^{k-3}(1-t)^{N-k-1} dt
$$

$$
- \frac{(N-1)!}{(k-1)!(N-k-1)!} \frac{2(k+1)}{N^2} \int_{t=0}^{1} t^{k-2}(1-t)^{N-k-1} dt
$$

$$
+ \frac{(N-1)!}{(k-1)!(N-k-1)!} \int_{t=0}^{1} t^{k-3}(1-t)^{N-k-1} dt
$$

$$
= \quad \frac{(N-1)!}{(k-1)!(N-k-1)!} \frac{(k+1)^2}{N^2} \frac{(k-3)!(N-k-1)!}{(N-3)!}
$$

$$
- \frac{(N-1)!}{(k-1)!(N-k-1)!} \frac{2(k+1)}{N^2} \frac{(k-2)!(N-k-1)!}{(N-2)!} + 1
$$

$$
= \quad \frac{(N-1)(N-2)(k+1)^2}{N^2(k-1)(k-2)} - \frac{2(N-1)(k+1)}{N(k-1)} + 1 . \quad (34)
$$

For sufficiently large $N$ and $k$, it is upper bounded by $C_1(1/N + 1/k)$ for some constant $C_1 > 0$. Therefore,

$$
\mathbb{E}_r \left[ \log \left( \frac{N(P_X(x,r) - P_{XY}(x,y,r)) + (k+1)(1 - P_X(x,r))}{NP_X(x,r)(1 - P_{XY}(x,y,r))} \right) \right] \leq \sqrt{C_1(\frac{1}{N} + \frac{1}{k})} . \quad (35)
$$

Similarly, by using the fact that $\log(x/y) > (x - y)/x$ and Cauchy-Schwarz inequality again, we conclude that there are some constant $C_2 > 0$ such that

$$
\mathbb{E}_r \left[ \log \left( \frac{N(P_X(x,r) - P_{XY}(x,y,r)) + (k+1)(1 - P_X(x,r))}{NP_X(x,r)(1 - P_{XY}(x,y,r))} \right) \right] \geq -\sqrt{C_2(\frac{1}{N} + \frac{1}{k})} . \quad (36)
$$

Therefore, by combining (32), (35) and (36), we obtain

$$
\left| \int_{r=0}^{\infty} \left( \mathbb{E} \left[ \log(n_{x,1} + 1) | (X,Y) = (x,y), \rho_{k,1} = r \right] - \log N - \log P_X(x,r) \right) dF_{\rho_{k,1}}(r) \right|
$$

$$
\leq \quad \frac{C}{k+1} + \sqrt{C'(\frac{1}{N} + \frac{1}{k})} . \quad (37)
$$

where $C' = \max\{C_1, C_2\}$. Since (24) and (23) are symmetric, the same upper bound (37) also applies to (24). Combine (28), (29) and (37), we have

$$\left| \mathbb{E}\left[\xi_1|(X,Y)=(x,y)\right] - \log f(x,y) \right|$$

$$\leq \delta_N + \frac{k(3\log N + |\log f(x,y)|)}{N^k} + \log N - \psi(N) + \frac{2C}{k+1} + 2\sqrt{C'(\frac{1}{N} + \frac{1}{k})} \quad (38)$$

for every $(x,y) \in \Omega_3 \setminus E$. By integration over $\Omega_3 \setminus E$, we have

$$\int_{\Omega_3 \setminus E} \left| \mathbb{E}\left[\xi_1|(X,Y)=(x,y)\right] - \log f(x,y) \right| dP_{XY}$$

$$\leq \int_{\Omega_3 \setminus E} \left( \delta_N + \frac{k(3\log N + |\log f(x,y)|)}{N^k} + \log N - \psi(N) \right.$$

$$\left. + \frac{2C}{k+1} + 2\sqrt{C'(\frac{1}{N} + \frac{1}{k})} \right) dP_{XY}$$

$$\leq \delta_N + \frac{k(3\log N + \int_{\mathcal{X} \times \mathcal{Y}} |\log f(x,y)| dP_{XY})}{N^k} + \log N - \psi(N)$$

$$+ \frac{2C}{k+1} + 2\sqrt{C'(\frac{1}{N} + \frac{1}{k})} . \quad (39)$$

By Assumption 1, $k$ increases as $N \to \infty$. By Assumption 5, $\int_{\mathcal{X} \times \mathcal{Y}} |\log f(x,y)| dP_{XY} < +\infty$. Therefore, this quantity vanishes as $N \to \infty$. Combining with the case that $(x,y) \in E$, we have

$$\lim_{N \to \infty} \int_{\Omega_3} \left| \mathbb{E}\left[\xi_1|(X,Y)=(x,y)\right] - \log f(x,y) \right| dP_{XY} = 0 \quad (40)$$

## B.1 Proof of Lemma B.1

We will need to prove that for any measurable set $A \subseteq \mathcal{X} \times \mathcal{Y}$, we have $\int_A f dP_X P_Y = P_{XY}(A)$. For any $\epsilon > 0$, by Egoroff's Theorem, there exists $B \subseteq \mathcal{X} \times \mathcal{Y}$ such that $P_{XY}(B^C) < \epsilon$, $P_X P_Y(B^C) < \epsilon$ and $P_{XY}(x,y,r)/P_X(x,r)P_Y(y,r)$ converges to $f(x,y)$ uniformly on $B$. Now we have:

$$|P_{XY}(A) - \int_A f dP_X P_Y|$$

$$= |P_{XY}(A \cap B) + P_{XY}(A \cap B^C) - \int_{A \cap B} f dP_X P_Y - \int_{A \cap B^C} f dP_X P_Y|$$

$$\leq |P_{XY}(A \cap B) - \int_{A \cap B} f dP_X P_Y| + P_{XY}(A \cap B^C) + \int_{A \cap B^C} f dP_X P_Y$$

$$\leq |P_{XY}(A \cap B) - \int_{A \cap B} f dP_X P_Y| + P_{XY}(B^C) + C P_X P_Y(B^C)$$

$$\leq |P_{XY}(A \cap B) - \int_{A \cap B} f dP_X P_Y| + \epsilon(1 + C), \quad (41)$$

where $C$ is the upper bound for $f(x,y)$ in Assumption 3. Now we need to deal with the first term of (41). By Assumption 4, $\mathcal{X} \times \mathcal{Y}$ can be decomposed into countable disjoint sets $\{E_i\}_{i=1}^\infty$ such that $f(x,y)$ is uniformly continuous on each $E_i$, so by define $A_i = A \cap B \cap E_i$, we have

$$|P_{XY}(A \cap B) - \int_{A \cap B} f dP_X P_Y| \leq \sum_{i=1}^\infty |P_{XY}(A_i) - \int_{A_i} f dP_X P_Y|. \quad (42)$$

Since $f(x,y)$ is uniformly continuous on $E_i$, so there exists $\delta_1 > 0$ such that for every $(x_1, y_1) \in A_i \subseteq E_i$ and $(x_2, y_2) \in A_i \subseteq E_i$ such that $\|x_1 - x_2\| < \delta_1$ and $\|y_1 - y_2\| < \delta_1$, we have $|f(x_1, y_1) - f(x_2, y_2)| < \epsilon$. Additionally, since $P_{XY}(x,y,r)/P_X(x,r)P_Y(y,r)$ converges to $f(x,y)$ uniformly on $B$, there exists $\delta_2 > 0$ such that for every $(x,y) \in A_i \subseteq B$ and $r < \delta_2$, we have $|P_X Y(x,y,r)/P_X(x,r)P_Y(y,r) - f(x,y)| < \epsilon$. Take $\delta = \min\{\delta_1, \delta_2\}$. Since $A_i$ is a subset

of Euclidean space, we can decompose $A_i$ as $A_i = \cup_{j=1}^{\infty} A_{ij}$, where $A_{ij}$ is a square set centered at $(x_{ij}, y_{ij})$ with radius $r_{ij} < \delta$. Then consider the following simple function $\phi(x, y)$,

$$\phi(x, y) \equiv \begin{cases} \frac{P_{XY}(A_{ij})}{P_X(A_i)P_Y(A_i)} = \frac{P_{XY}(x_{ij}, y_{ij}, r_{ij})}{P_X(x_{ij}, r_{ij})P_Y(y_{ij}, r_{ij})}, & \text{if } (x, y) \in A_{i,j} \\ 0, & \text{otherwise} \end{cases}. \tag{43}$$

Then we have

$$\int_{A_i} \phi(x, y) dP_X P_Y = \sum_{j=1}^{\infty} \int_{A_{ij}} \frac{P_{XY}(A_{ij})}{P_X(A_i)P_Y(A_i)} dP_X P_Y = \sum_{j=1}^{\infty} P_{XY}(A_{ij}) = P_{XY}(A_i) \tag{44}$$

and

$$|\phi(x, y) - f(x, y)| \leq |\frac{P_{XY}(x_{ij}, y_{ij}, r_{ij})}{P_X(x_{ij}, r_{ij})P_Y(y_{ij}, r_{ij})} - f(x_{ij}, y_{ij})| + |f(x_{ij}, y_{ij}) - f(x, y)|$$

$$\leq \epsilon + \epsilon = 2\epsilon \tag{45}$$

for every $(x, y) \in A_{ij}$. Therefore, we have

$$|P_{XY}(A_i) - \int_{A_i} f dP_X P_Y| = |\int_{A_i} \phi dP_X P_Y - \int_{A_i} f dP_X P_Y|$$

$$\leq \int_{A_i} |\phi - f| dP_X P_Y \leq 2\epsilon P_X P_Y(A_i). \tag{46}$$

Plug this to (42), we have:

$$|P_{XY}(A \cap B) - \int_{A \cap B} f dP_X P_Y| \leq \sum_{i=1}^{\infty} 2\epsilon P_X P_Y(A_i) = 2\epsilon P_X P_Y(\bigcup_{i=1}^{\infty} A_i) \leq 2\epsilon. \tag{47}$$

Plug this to (41), we have $|P_{XY}(A) - \int_A f dP_X P_Y| < (3 + C)\epsilon$. Notice that this statement holds for any $\epsilon > 0$. By choosing $\epsilon \downarrow 0$, we conclude that $P_{XY}(A) = \int_A f dP_X P_Y$. Hence, $f$ is the Radon-Nikolym derivative.

## B.2 Proof of Lemma B.2

Given that $(X_1, Y_1) = (x, y)$ and $\rho_{k,1} = r$, we sort the samples $\{(X_i, Y_i)\}_{i=2}^N$ by their distance to $(x, y)$ defined as $d_i = \max\{\|X_i - x\|, \|Y_i - y\|\}$. To avoid the case that two samples have identical distance, we introduce a set of random variables $\{Z_i\}_{i=2}^N$ i.i.d. samples from $\text{Unif}[0, 1]$ and define a comparison operator $\prec$ as:

$$i \prec j \iff d_i < d_j \quad \text{or} \quad \{d_i = d_j \quad \text{and} \quad Z_i < Z_j\}. \tag{48}$$

Since for any $i \neq j$, the probability that $Z_i = Z_j$ is zero, so we can have either $i \prec j$ or $i \succ j$ with probability 1. Now let $\{2, 3, \ldots, N\} = S \cup \{j\} \cup T$ be a partition of the indices with $|S| = k - 1$ and $|T| = N - k - 1$. Define an event $\mathcal{A}_{S,j,T}$ associated to the partition as:

$$\mathcal{A}_{S,j,T} = \{ s \prec j, \forall s \in S, \text{ and } t \succ j, \forall t \in T \}. \tag{49}$$

Since $(X_j, Y_j) - (x, y)$ are i.i.d. random variables each of the events $\mathcal{A}_{S,j,T}$ has identical probability. The number of all partitions is $\frac{(N-1)!}{(N-k-1)!(k-1)!}$ and thus $\mathbb{P}(\mathcal{A}_{S,j,T}) = \frac{(N-k-1)!(k-1)!}{(N-1)!}$. So the cdf of $\tilde{k}_1$ is given by:

$$\mathbb{P}\left( \tilde{k}_1 \leq k + m | \rho_{k,1} = r, (X_1, Y_1) = (x, y) \right)$$

$$= \sum_{S,j,T} \mathbb{P}(\mathcal{A}_{S,j,T} | \rho_{k,1} = r, (X_1, Y_1) = (x, y)) \mathbb{P}\left( \tilde{k}_1 \leq k + m | \mathcal{A}_{S,j,T}, \rho_{k,1} = r, (X_1, Y_1) = (x, y) \right)$$

$$= \frac{(N - k - 1)!(k - 1)!}{(N - 1)!} \sum_{S,j,T} \mathbb{P}\left( \tilde{k}_1 \leq k + m | \mathcal{A}_{S,j,T}, \rho_{k,1} = r, (X_1, Y_1) = (x, y) \right) \tag{50}$$

Now condition on event $\mathcal{A}_{S,j,T}$ and $\rho_{k,1} = r$, namely $(X_j, Y_j)$ is the $k$-nearest neighbor with distance $r$, $S$ is the set of samples with distance smaller than (or equal to) $r$ and $T$ is the set of samples with

distance greater than (or equal to) $r$. Recall that $\tilde{k}_1$ is the number of samples with $d_j \leq r$. For any index $s \in S \cup \{j\}$, $d_j \leq r$ are satisfied. Therefore, $\tilde{k}_1 \leq k + m$ means that there are no more than $m$ samples in $T$ with distance smaller than $r$. Let $U_l = \mathbb{I}\{d_l \leq r \,|\, d_l \geq r\}$. Therefore,

$$\mathbb{P}\left( \tilde{k}_1 \leq k + m \big| \mathcal{A}_{S,j,T}, \rho_{k,1} = r, (X_1, Y_1) = (x, y) \right)$$

$$= \mathbb{P}\left( \sum_{l \in T} \mathbb{I}\{d_l \leq r\} \leq m \,\big|\, d_s \leq r, \forall s \in S, d_j = r, d_t \geq r, \forall t \in T \right)$$

$$= \mathbb{P}\left( \sum_{l \in T} \mathbb{I}\{d_l \leq r\} \leq m \,\big|\, d_l \geq r, \forall l \in T \right) = \mathbb{P}\left( \sum_{l \in T} U_l \leq m \right), \quad (51)$$

where $U_l$ follows bernoulli distribution with $\mathbb{P}\{U_l = 1\} = Pr\{d_l \leq r | d_l \geq r\}$. We can drop the conditioning of $(X_s, Y_s)$'s for $s \notin T$ since $(X_s, Y_s)$ and $(X_t, Y_t)$ are independent. Therefore, given that $d_l \geq r$ for all $l \in T$, the variables $\mathbb{I}\{d_l \leq r\}$ are i.i.d. and have the same distribution as $U_l$. We conclude:

$$\mathbb{P}\left( \tilde{k}_1 \leq k + m \big| \rho_{k,1} = r, (X_1, Y_1) = (x, y) \right)$$

$$= \frac{(N - k - 1)!(k - 1)!}{(N - 1)!} \sum_{S,j,T} \mathbb{P}\left( \tilde{k}_1 \leq k + m \big| \mathcal{A}_{S,j,T}, \rho_{i,xy} = r, (X_1, Y_1) = (x, y) \right)$$

$$= \frac{(N - k - 1)!(k - 1)!}{(N - 1)!} \sum_{S,j,T} \mathbb{P}\left( \sum_{l \in T} U_l \leq m \right) = \mathbb{P}\left( \sum_{l \in T} U_l \leq m \right). \quad (52)$$

Thus we have shown that $\tilde{k}_i - k$ has the same distribution as $\sum_{l \in T} U_l$, which is a Binomial random variable with parameter $|T| = N - k - 1$ and $\mathbb{P}\{d_l \leq r \,|\, d_l \geq r\} = \mathbb{P}\{d_l = 0\} = P_{XY}(x, y, 0)$. For $n_{x,1}$ and $n_{y,1}$, we can follow the same proof and conclude that $n_{x,i} - k$ and $n_{y,i} - k$ are also Binomial random variables with $|T| = N - k - 1$. But the probabilities are different.

- If $r = 0$, then for $n_{x,i}$, the probability is $\mathbb{P}\{\|X_l - x\| \leq 0 \,|\, d_l \geq 0\} = \mathbb{P}\{\|X_l - x\| = 0\} = P_X(x, 0)$ and the probability for $n_{y,i}$ is $P_Y(y, 0)$.

- If $r > 0$, then for $n_{x,i}$, the probability is $\mathbb{P}\{\|X_l - x\| \leq r \,|\, d_l \geq r\} = \frac{P_X(x,r) - P_{XY}(x,y,r)}{1 - P_{XY}(x,y,r)}$. Similarly, the probability for $n_{y,i}$ is $\frac{P_Y(x,r) - P_{XY}(x,y,r)}{1 - P_{XY}(x,y,r)}$.

### B.3 Proof of Lemma B.3

By Jensen's inequality, we know that $\mathbb{E}[\log X] \leq \log \mathbb{E}[X] = \log(Np + k)$. So it suffices to give an upper bound for $\log(Np + k) - \mathbb{E}[\log X]$. We consider two different cases.

(i) $Np \geq k$. In this case, for any $x$, by applying Taylor's theorem around $x_0 = Np + k$, there exists $\zeta$ between $x$ and $x_0$ such that

$$\log(x) = \log(Np + k) + \frac{x - Np - k}{Np + k} - \frac{(x - Np - k)^2}{2\zeta^2} \quad (53)$$

By noticing that $\zeta \geq \min\{x, x_0\} = \min\{x, Np + k\}$, we have

$$-\log(x) + \log(Np + k) + \frac{x - Np - k}{Np + k} = \frac{(x - Np - k)^2}{2\zeta^2}$$

$$\leq \max\left\{\frac{(x - Np - k)^2}{2x^2}, \frac{(x - Np - k)^2}{2(Np + k)^2}\right\} \leq \frac{(x - Np - k)^2}{2x^2} + \frac{(x - Np - k)^2}{2(Np + k)^2}. \quad (54)$$

Now let $X - k$ be a $\text{Bino}(N, p)$ random variable. By taking expectation on both sides, we have:

$$-\mathbb{E}[\log X] + \log(Np + k) + \frac{\mathbb{E}[X] - Np - k}{Np + k}$$

$$\leq \mathbb{E}\left[\frac{(X - Np - k)^2}{2X^2}\right] + \frac{\mathbb{E}\left[(X - Np - k)^2\right]}{2(Np + k)^2}. \quad (55)$$

Since $\mathbb{E}[X] = Np + k$, $\mathbb{E}\left[(X - Np - k)^2\right] = \text{Var}\left[X\right] = Np(1 - p)$, and

$$
\begin{aligned}
\mathbb{E}\left[\frac{(X - Np - k)^2}{2X^2}\right] &= \sum_{j=0}^{N} \frac{(j - Np)^2}{2(j + k)^2} \binom{N}{j} p^j (1 - p)^{N-j} \\
&\leq \sum_{j=0}^{N} \frac{(j - Np)^2}{2(j + 2)(j + 1)} \binom{N}{j} p^j (1 - p)^{N-j} \\
&= \sum_{j=0}^{N} \frac{(j - Np)^2}{2(N + 2)(N + 1)p^2} \binom{N + 2}{j + 2} p^{j+2} (1 - p)^{N-j} \\
&\leq \frac{1}{2(N + 2)(N + 1)p^2} \mathbb{E}_{Y \sim \text{Bino}(N+2,p)}\left[(Y - Np)^2\right] \\
&= \frac{(N + 2)p(1 - p) + 4p^2}{2(N + 2)(N + 1)p} \leq \frac{(N + 2)p}{2(N + 2)(N + 1)p} \leq \frac{1}{2Np} \quad (56)
\end{aligned}
$$

for $k \geq 2$ and $N \geq 4$. Plug these in (55), we have

$$
\begin{aligned}
-\mathbb{E}[\log X] + \log(Np + k) &\leq \frac{1}{2Np} + \frac{Np(1 - p)}{2(Np + k)^2} \\
&\leq \frac{1}{Np + k} + \frac{1}{2(Np + k)} = \frac{3}{2(Np + k)} . \quad (57)
\end{aligned}
$$

where $1/(2Np) \leq 1/(Np + k)$ comes from the fact that $Np \geq k$.

(ii) $Np < k$. In this case, for any $x$, by applying Taylor's theorem around $x_0 = Np + k$, there exists $\zeta$ between $x$ and $x_0$ such that

$$
\log(x) = \log(Np + k) + \frac{x - Np - k}{Np + k} - \frac{(x - Np - k)^2}{2\zeta^2} \quad (58)
$$

By noticing that $\zeta \geq \min\{x, x_0\} \geq k \geq (Np + k)/2$, we have:

$$
-\log(x) + \log(Np + k) + \frac{x - Np - k}{Np + k} \leq \frac{2(x - Np - k)^2}{(Np + k)^2} . \quad (59)
$$

Similarly, by taking expectation on both sides, we have

$$
-\mathbb{E}[\log X] + \log(Np + k) + \frac{\mathbb{E}[X] - Np - k}{Np + k} \leq \frac{\mathbb{E}\left[2(X - Np - k)^2\right]}{(Np + k)^2} . \quad (60)
$$

By plugging in $\mathbb{E}[X] = Np + k$ and $\mathbb{E}\left[(X - Np - k)^2\right] = \text{Var}\left[X\right] = Np(1 - p)$, we obtain

$$
-\mathbb{E}[\log X] + \log(Np + k) \leq \frac{2Np(1 - p)}{(Np + k)^2} \leq \frac{2(Np + k)}{(Np + k)^2} = \frac{2}{Np + k} . \quad (61)
$$

Combining the two cases, we obtain the desired statement.

## C   Proof of Theorem 2

We use the Efron-Stein inequality to bound the variance of the estimator. For simplicity, let $\widehat{I}^{(N)}(Z)$ be the estimate based on original samples $\{Z_1, Z_2, \ldots, Z_N\}$, where $Z_i = (X_i, Y_i)$. For the usage of Efron-Stein inequality, we consider another set of i.i.d. samples $\{Z_1', Z_2', \ldots, Z_n'\}$ drawn from $P_{XY}$. Let $\widehat{I}^{(N)}(Z^{(j)})$ be the estimate based on $\{Z_1, \ldots, Z_{j-1}, Z_j', Z_{j+1}, \ldots, Z_N\}$. Then Efron-Stein inequality states that

$$
\text{Var}\left[\widehat{I}^{(N)}(Z)\right] \leq \frac{1}{2} \sum_{j=1}^{N} \mathbb{E}\left[\left(\widehat{I}^{(N)}(Z) - \widehat{I}^{(N)}(Z^{(j)})\right)^2\right] . \quad (62)
$$

Now we will give an upper bound for the difference $|\widehat{I}^{(N)}(Z) - \widehat{I}^{(N)}(Z^{(j)})|$ for given index $j$. First of all, let $\widehat{I}^{(N)}(Z_{\setminus j})$ be the estimate based on $\{Z_1, \ldots, Z_{j-1}, Z_{j+1}, \ldots, Z_N\}$, then by triangle inequality, we have:

$$\sup_{Z_1, \ldots, Z_N, Z_j'} \left| \widehat{I}^{(N)}(Z) - \widehat{I}^{(N)}(Z^{(j)}) \right|$$

$$\leq \sup_{Z_1, \ldots, Z_N, Z_j'} \left( \left| \widehat{I}^{(N)}(Z) - \widehat{I}^{(N)}(Z_{\setminus j}) \right| + \left| \widehat{I}^{(N)}(Z_{\setminus j}) - \widehat{I}^{(N)}(Z^{(j)}) \right| \right)$$

$$\leq \sup_{Z_1, \ldots, Z_N} \left| \widehat{I}^{(N)}(Z) - \widehat{I}^{(N)}(Z_{\setminus j}) \right| + \sup_{Z_1, \ldots, Z_{j-1}, Z_j', Z_{j+1}, \ldots, Z_N} \left| \widehat{I}^{(N)}(Z_{\setminus j}) - \widehat{I}^{(N)}(Z^{(j)}) \right|$$

$$= 2 \sup_{Z_1, \ldots, Z_N} \left| \widehat{I}^{(N)}(Z) - \widehat{I}^{(N)}(Z_{\setminus j}) \right| \tag{63}$$

where the last equality comes from the fact that $\{Z_1, \ldots, Z_{j-1}, Z_j', Z_{j+1}, \ldots, Z_N\}$ has the same joint distribution as $\{Z_1, \ldots, Z_N\}$. Now recall that

$$\widehat{I}^{(N)}(Z) = \frac{1}{N} \sum_{i=1}^{N} \xi_i(Z) = \frac{1}{N} \sum_{i=1}^{N} \left( \psi(\tilde{k}_i) + \log N - \log(n_{x,i} + 1) - \log(n_{y,i} + 1) \right), \tag{64}$$

Therefore, we have

$$\sup_{Z_1, \ldots, Z_N, Z_j'} \left| \widehat{I}^{(N)}(Z) - \widehat{I}^{(N)}(Z^{(j)}) \right| \leq \frac{2}{N} \sup_{Z_1, \ldots, Z_N} \sum_{i=1}^{N} \left| \xi_i(Z) - \xi_i(Z_{\setminus j}) \right|. \tag{65}$$

Now we need to upper-bound the difference $|\xi_i(Z) - \xi_i(Z_{\setminus j})|$ created by eliminating sample $Z_j$ for different $i$'s. There are three cases of $i$'s as follows,

- **Case I.** $i = j$. Since the upper bounds $|\xi_i(Z)| \leq 2\log N$ and $|\xi_i(Z_{\setminus j})| \leq 2\log(N-1)$ always holds, so $|\xi_i(Z) - \xi_i(Z_{\setminus j})| \leq 4\log N$. The number of $i$'s in this case is only 1. So $\sum_{\text{Case I}} |\xi_i(Z) - \xi_i(Z_{\setminus j})| \leq 4\log N$.

- **Case II.** $\rho_{i,xy} = 0$. In this case, recall that $\tilde{k}_i = \left| \{ i' \neq i : Z_i = Z_{i'} \} \right|$, $n_{x,i} = \left| \{ i' \neq i : X_i = X_{i'} \} \right|$ and $n_{y,i} = \left| \{ i' \neq i : Y_i = Y_{i'} \} \right|$. There are 4 sub-cases in this case.

  - **Case II.1.** $Z_i = Z_j$. By eliminating $Z_j$, $\tilde{k}_i, n_{x,i}, n_{y,i}$ will all decrease by 1. Therefore,

$$|\xi_i(Z) - \xi_i(Z_{\setminus j})|$$

$$= \left| \left( \psi(\tilde{k}_i) + \log N - \log(n_{x,i} + 1) - \log(n_{y,i} + 1) \right) \right.$$

$$\left. - \left( \psi(\tilde{k}_i - 1) + \log(N - 1) - \log(n_{x,i}) - \log(n_{y,i}) \right) \right|$$

$$\leq |\psi(\tilde{k}_i) - \psi(\tilde{k}_i - 1)| + |\log N - \log(N - 1)|$$

$$+ |\log(n_{x,i} + 1) - \log(n_{x,i})| + |\log(n_{y,i} + 1) - \log(n_{y,i})|$$

$$\leq \frac{1}{\tilde{k}_i - 1} + \frac{1}{N - 1} + \frac{1}{n_{x,i}} + \frac{1}{n_{y,i}} \leq \frac{4}{\tilde{k}_i - 1} = \frac{4}{\tilde{k}_j - 1}. \tag{66}$$

  The number of $i$'s in this case is the number if $i$'s such that $Z_i = Z_j$, which is just $\tilde{k}_j$. Therefore, $\sum_{\text{Case II.1}} |\xi_i(Z) - \xi_i(Z_{\setminus j})| \leq 4\tilde{k}_j/(\tilde{k}_j - 1) \leq 8$, for $\tilde{k}_j \geq k \geq 2$.

  - **Case II.2.** $X_i = X_j$ but $Y_i \neq Y_j$. By eliminating $Z_j$, $\tilde{k}_i$ and $n_{y,i}$ won't change but $n_{x,i}$ will decrease by 1. Therefore,

$$|\xi_i(Z) - \xi_i(Z_{\setminus j})| \leq |\log N - \log(N - 1)| + |\log(n_{x,i} + 1) - \log(n_{x,i})|$$

$$\leq \frac{1}{N - 1} + \frac{1}{n_{x,i}} \leq \frac{2}{n_{x,i}} = \frac{2}{n_{x,j}} \tag{67}$$

  The number of $i$'s in this case is the number if $i$'s such that $X_i = X_j$ but $Y_i \neq Y_j$, which is less than $n_{x,j}$. Therefore, $\sum_{\text{Case II.2}} |\xi_i(Z) - \xi_i(Z_{\setminus j})| \leq 2n_{x,j}/n_{x,j} \leq 2$.

- **Case II.3.** $Y_i = Y_j$ but $X_i \neq X_j$. By eliminating $Z_j$, $\tilde{k}_i$ and $n_{x,i}$ won't change but $n_{y,i}$ will decrease by 1. Similarly as Case II.2, we have $\sum_{\text{Case II.3}} |\xi_i(Z) - \xi_i(Z_{\setminus j})| \leq 2$.

- **Case II.4.** $X_i \neq X_j$ and $Y_i \neq Y_j$. In this case, none of $\tilde{k}_i$, $n_{x,i}$, or $n_{y,i}$ will change. So $|\xi_i(Z) - \xi_i(Z_{\setminus j})| = \log N - \log(N-1) \leq 1/(N-1)$. The number of $i$'s in this case is simply less than $N-1$. Therefore, $\sum_{\text{Case II.4}} |\xi_i(Z) - \xi_i(Z_{\setminus j})| \leq 1$.

Combining the four sub-cases, we conclude that $\sum_{\text{Case II}} |\xi_i(Z) - \xi_i(Z_{\setminus j})| \leq 13$.

- **Case III.** $\rho_{i,xy} > 0$. In this case, recall that $\tilde{k}_i$ always equals to $k$, $n_{x,i} = \left| \{i' \neq i : \|X_i - X_{i'}\| \leq \rho_{i,xy}\} \right|$ and $n_{y,i} = \left| \{i' \neq i : \|Y_i - Y_{i'}\| \leq \rho_{i,xy}\} \right|$. Similar to Case II, there are 4 sub-cases.

  - **Case III.1.** $Z_j$ is in the $k$-nearest neighbors of $Z_i$. In this case, we don't know how $n_{x,i}$ and $n_{y,i}$ will change by eliminating $Z_j$, so we just use the loosest bound $|\xi_i(Z) - \xi_i(Z_{\setminus j})| \leq 4 \log N$. However, the number of $i$'s in this case is upper bounded by the following lemma.

    **Lemma C.1.** *Let $Z, Z_1, Z_2, \ldots, Z_N$ be vectors of $\mathbb{R}^d$ and $\mathcal{Z}_i$ be the set $\{Z_1, \ldots, Z_{i-1}, Z, Z_{i+1}, \ldots, Z_N\}$. Then*

    $$\sum_{i=1}^{N} \mathbb{I}\{Z \text{ is in the } k\text{-nearest neighbors of } Z_i \text{ in } \mathcal{Z}_i\} \leq k\gamma_d , \qquad (68)$$

    *(distance ties are broken by comparing indices). Here $\gamma_d$ is the minimum number of cones with angle smaller than $\pi/6$ needed to cover $\mathbb{R}^d$. Moreover, if we allow $k$ to be different for difference $i$, we have*

    $$\sum_{i=1}^{N} \frac{1}{k_i} \mathbb{I}\{Z \text{ is in the } k_i\text{-nearest neighbors of } Z_i \text{ in } \mathcal{Z}_i\} \leq \gamma_d(\log N + 1) . \qquad (69)$$

    By the first inequality in Lemma C.1, the number of $i$'s in this case is upper bounded by $k\gamma_d$. Therefore, $\sum_{\text{Case III.1}} |\xi_i(Z) - \xi_i(Z_{\setminus j})| \leq 4k\gamma_{d_x+d_y} \log N$.

  - **Case III.2.** $Z_j$ is not in the $k$-nearest neighbors of $Z_i$, but $\|X_j - X_i\| \leq \rho_{i,xy}$, i.e., $X_j$ is in the $n_{x,i}$-nearest neighbors of $X_i$. In this case, $n_{x,i}$ will decrease by 1 and $n_{y,i}$ remains the same. So

    $$\begin{aligned} |\xi_i(Z) - \xi_i(Z_{\setminus j})| &\leq |\log N - \log(N-1)| + |\log(n_{x,i}+1) - \log(n_{x,i})| \\ &\leq \frac{1}{N-1} + \frac{1}{n_{x,i}} \leq \frac{2}{n_{x,i}} \end{aligned} \qquad (70)$$

    We don't have an upper bound for the number of $i$'s in this case, but from the second inequality in Lemma C.1, we have the following upper bound, where $\mathcal{X}_{i,j} = \{X_1, \ldots, X_{i-1}, X_j, X_{i+1}, \ldots, X_N\}$:

    $$\begin{aligned} \sum_{\text{Case III.2}} & |\xi_i(Z) - \xi_i(Z_{\setminus j})| \\ &\leq \sum_{i=1}^{N} \frac{2}{n_{x,i}} \mathbb{I}\{X_j \text{ is in the } n_{x,i}\text{-nearest neighbors of } X_i \text{ in } \mathcal{X}_{i,j}\} \\ &\leq 2\gamma_{d_x}(\log N + 1) \leq 2\gamma_{d_x+d_y}(\log N + 1) . \end{aligned} \qquad (71)$$

  - **Case III.3.** $Z_j$ is not in the $k$-nearest neighbors of $Z_i$, but $\|Y_j - Y_i\| \leq \rho_{i,xy}$, i.e., $Y_j$ is in the $n_{y,i}$-nearest neighbors of $Y_i$. In this case, $n_{y,i}$ will decrease by 1 and $n_{x,i}$ remains the same. Follow the same analysis in Case III.2, we have $\sum_{\text{Case III.2}} |\xi_i(Z) - \xi_i(Z_{\setminus j})| \leq 2\gamma_{d_x+d_y}(\log N + 1)$ as well.

  - **Case III.4.** $Z_j$ is not in the $k$-nearest neighbors of $Z_i$, and $\|X_j - X_i\| > \rho_{i,xy}$, $\|Y_j - Y_i\| > \rho_{i,xy}$. In this case, neither $n_{x,i}$ nor $n_{y,i}$ will change. Similar to Case II.4, $\sum_{\text{Case III.4}} |\xi_i(Z) - \xi_i(Z_{\setminus j})| \leq 1$.

Combining the four sub-cases, we conclude that $\sum_{\text{Case III}} |\xi_i(Z) - \xi_i(Z_{\setminus j})| \leq (4k + 4)\gamma_{d_x+d_y} \log N + 4\gamma_{d_x+d_y} + 1$.

Combining the three cases, we have:

$$\sum_{i=1}^{N} \left| \xi_i(Z) - \xi_i(Z_{\setminus j}) \right| \leq 4 \log N + 13 + (4k+4)\gamma_{d_x+d_y} \log N + 4\gamma_{d_x+d_y} + 1$$

$$\leq 30 \gamma_{d_x+d_y} k \log N \tag{72}$$

for $k \geq 1$, $\log N \geq 1$ and all $\{Z_1, \ldots, Z_N\}$. Plug it into (65), we obtain,

$$\sup_{Z_1, \ldots, Z_N, Z'_j} \left| \widehat{I}^{(N)}(Z) - \widehat{I}^{(N)}(Z^{(j)}) \right| \leq \frac{60 \gamma_{d_x+d_y} k \log N}{N} . \tag{73}$$

Plug it into Efron-Stein inequality (62), we obtain:

$$\begin{aligned}
\text{Var}\left[ \widehat{I}^{(N)}(Z) \right] &\leq \frac{1}{2} \sum_{j=1}^{N} \mathbb{E}\left[ \left( \widehat{I}^{(N)}(Z) - \widehat{I}^{(N)}(Z^{(j)}) \right)^2 \right] \\
&\leq \frac{1}{2} \sum_{j=1}^{N} \sup_{Z_1, \ldots, Z_n, Z'_j} \left( \widehat{I}^{(N)}(Z) - \widehat{I}^{(N)}(Z^{(j)}) \right)^2 \\
&\leq \frac{1}{2} \sum_{j=1}^{N} (\frac{60 \gamma_{d_x+d_y} k \log N}{N})^2 = \frac{1800 \gamma_{d_x+d_y}^2 (k \log N)^2}{N} .
\end{aligned} \tag{74}$$

Since $1800 \gamma_{d_x+d_y}^2$ is a constant independent of $N$, and $(k_N \log N)^2/N \to 0$ as $N \to \infty$ by Assumption 6, we have $\lim_{N \to \infty} \text{Var}\left[ \widehat{I}^{(N)}(Z) \right] = 0$.

## C.1 Proof of Lemma C.1

For the first part of the lemma, we refer to Lemma 20.6 in [5].

The second part of the lemma is a consequence of the first part. We reorder the indices $i$'s by $k_i$ and rewrite the summation as follows,

$$\begin{aligned}
&\sum_{i=1}^{N} \frac{1}{k_i} \mathbb{I}\{Z \text{ is in the } k_i\text{-nearest neighbors of } Z_i \text{ in } \mathscr{Z}_i\} \\
= &\sum_{k=1}^{N} \frac{1}{k} \sum_{i=1}^{N} \mathbb{I}\{k_i = k\} \mathbb{I}\{Z \text{ is in the } k\text{-nearest neighbors of } Z_i \text{ in } \mathscr{Z}_i\} \\
= &\sum_{k=1}^{N} \frac{1}{k} \sum_{i=1}^{N} \mathbb{I}\{k_i = k \text{ and } Z \text{ is in the } k\text{-nearest neighbors of } Z_i \text{ in } \mathscr{Z}_i\}
\end{aligned} \tag{75}$$

Notice that we take the summation over $k = 1$ to $N$ since each $k_i$ can not be more than $N$. Denote $S_k = \sum_{i=1}^{N} \mathbb{I}\{k_i = k \text{ and } Z \text{ is in the } k\text{-nearest neighbors of } Z_i \text{ in } \{Z_1, \ldots, Z_{i-1}, Z, Z_{i+1}, \ldots, Z_N\}\}$ for simplicity. Then we need to prove that $\sum_{k=1}^{N}(S_k/k) \leq \gamma_d \log N$. By the first part of this lemma, we obtain,

$$\begin{aligned}
\sum_{\ell=1}^{k} S_\ell &= \sum_{\ell=1}^{k} \sum_{i=1}^{N} \mathbb{I}\{k_i = \ell \text{ and } Z \text{ is in the } \ell\text{-nearest neighbors of } Z_i \text{ in } \mathscr{Z}_i\} \\
&= \sum_{i=1}^{N} \sum_{\ell=1}^{k} \mathbb{I}\{k_i = \ell \text{ and } Z \text{ is in the } \ell\text{-nearest neighbors of } Z_i \text{ in } \mathscr{Z}_i\} \\
&\leq \sum_{i=1}^{N} \mathbb{I}\{k_i \leq k \text{ and } Z \text{ is in the } k\text{-nearest neighbors of } Z_i \text{ in } \mathscr{Z}_i\} \\
&\leq k\gamma_d .
\end{aligned} \tag{76}$$

Therefore, we obtain

$$
\begin{aligned}
\sum_{k=1}^{N} \frac{S_k}{k} &= \sum_{k=1}^{N-1} \frac{1}{k(k+1)} \left( \sum_{\ell=1}^{k} S_\ell \right) + \frac{1}{N} \sum_{\ell=1}^{N} S_\ell \\
&\leq \sum_{k=1}^{N-1} \frac{k\gamma_d}{k(k+1)} + \frac{N\gamma_d}{N} = \sum_{k=1}^{N} \frac{\gamma_d}{k} < \gamma_d(\log N + 1) ,
\end{aligned}
\tag{77}
$$

which completes the proof.