[Reviews · NeurIPS 2017]

Reviewer 1



This paper deals with estimation of the mutual information of two measures under very weak assumptions, allowing for continuous, discrete and mixture of continuous and discrete distributions. The estimator is based on comparing the probability masses of Euclidean balls with identical radii for the three measures (namely, P_{XY}, P_X and P_Y), where the common radius is chosen based on the distance of the k_th nearest neighbor of the pair (X,Y). Consistency of the estimator is proven. It is argued that the estimator allows for more general distributions than previous attempts, especially those based on three differential entropies. The proofs are technical but seem to be correct and the paper is well written, offering a very elegant procedure under natural and general assumptions. I recommend its acceptance at NIPS.

Reviewer 2



Summary: This paper proposes an estimator for the mutual information of a pair (X,Y) of random variables given N joint samples. While most previous work has focused separately on the purely discrete and purely continuous methods (with significantly different methods used in each setting), the novel focus here on the case of combinations or mixtures or discrete and random variables; in this case, the mutual information is well-defined but cannot always be expressed in terms of entropy, making most previous estimators unusable. The proposed estimator generalizes the classic KSG estimator by defining the behavior at samples points with k-NN distance 0. The L_2 consistency of the estimator is proven, under a sort of continuity condition on the density ratio that allows the existence of atoms, and suitable scaling of k with N. Finally, experimental results on synthetic and real data compare the proposed estimator to some baseline estimators based on perturbing the data to make it purely discrete or continuous. Main Comments: This paper addresses a significant hole that remains in the nonparametric dependence estimation literature despite significant recent progress in this area; many current estimators are difficult or impossible to use with many data sets due to a combination or mixture of discrete and continuous components. While the proposed estimator seems like a fairly simple correction of the KSG estimator, and the consistency result is thus likely straightforward given the results of [16], the empirical section is strong and I think the work is of sufficient practical importance to merit acceptance. The paper is fairly well-written and easy to read. Minor Comments/questions: 1) As far as I understand 3H estimators can work as long as each individual dimension of the variables is fully discrete or full continuous, but fail if any variable is a mixture of the two (i.e., if an otherwise continuous distribution has any atoms). It might be good to clarify this, and also to compare to some 3H estimators (e.g., in experiment II). 2) Lines 39-42: Perhaps there is a typo or I am misreading something, but it isn't clear to my why the given definition of X isn't entirely discrete (it appears to be supported on {0,1,2,...}). 3) Lines 129-130: could be probably clearer "the average of Radon-Nikodym derivative" should be "the average of the logarithm Radon-Nikodym derivative of the joint distribution P_{XY} with respect to the product distribution P_XP_Y." 4) As discussed in Appendix A, a key observation of the paper is that, to be well-defined, mutual information requires only the trivial condition of absolute continuity of the joint distribution w.r.t. the marginal distribution (rather than w.r.t. another base measure, as for entropy). I think it would be worth explicitly mentioning this in the main paper, rather than just in the Appendix. 5) Although this paper doesn't study convergence rates, could the authors conjecture about how to quantify parameters that determine the convergence rate for mixed discrete/continuous variables? For continuous distributions, the smoothness of the density is usually relevant, while, for discrete distributions, the size of the support is usually relevant. However, mixture distributions are discontinuous and have infinite support. 6) Theorem 1 assumes that k -> infinity as N -> infinity. Can the authors conjecture whether the proposed estimator is consistent for fixed values of k? I ask because of recent work [4,16,43] showing that the KL and KSG estimators are consistent even for fixed k. 7) For some reason, the PDF is encoded such that I cannot search, highlight, or copy text. Perhaps it was converted from a postscript? Anyhow, this made the paper significantly more difficult to review. #### AFTER READING AUTHOR REBUTTAL #### "H(X,Y) is not well-defined as either discrete entropy or continuous entropy" - I think

Reviewer 3



Estimating mutual information has plenty of applications and have received renewed attention in recent years. In this paper, the authors focus on a particular case of estimating mutual information when the distributions are mixtures consisting of discrete and continuous components. The authors argue that this special case shows up in various applications, and point out that existing estimators work very badly in these cases, if they work at all. The proposed estimator works by identifying discrete points (by checking if the exact same point appeared k times) and uses a plug in estimator of the Radon-Nikodym derivative at these points. At continuous points, the estimator approximates the densities by considering how many samples lie within a ball of an appropriate radius (as suggested by the distance of the k-the nearest neighbour), and then estimates the Radon-Nikodym derivative at these point by using a KSG-like estimator. The two results are combined to arrive at the estimated mutual information. The authors show consistency of the proposed estimator under some conditions. The paper proposes a novel estimator, analyzes it theoretically, and also has compelling simulations.